# CROWDPLAY: CROWDSOURCING HUMAN DEMONSTRATIONS FOR OFFLINE LEARNING

**Matthias Gerstgrasser, Rakshit Trivedi & David C. Parkes**
School of Engineering and Applied Sciences
Harvard University
{matthias,rstrivedi,parkes}@seas.harvard.edu

## ABSTRACT

Crowdsourcing has been instrumental for driving AI advances that rely on large-scale data. At the same time, reinforcement learning has seen rapid progress through benchmark environments that strike a balance between tractability and real-world complexity, such as ALE and OpenAI Gym. In this paper, we aim to fill a gap at the intersection of these two: The use of crowdsourcing to generate large-scale human demonstration data in the support of advancing research into imitation learning and offline learning. To this end, we present *CrowdPlay*, a complete crowdsourcing pipeline for any standard RL environment including OpenAI Gym (made available under an open-source license); a large-scale publicly available crowdsourced dataset of human gameplay demonstrations in Atari 2600 games, including multimodal behavior and human-human and human-AI multiagent data; offline learning benchmarks with extensive human data evaluation; and a detailed study of incentives, including real-time feedback to drive high quality data. We hope that this will drive the improvement in design of algorithms that account for the complexity of human, behavioral data and thereby enable a step forward in direction of effective learning for real-world settings. Our code and dataset are available at https://mgerstgrasser.github.io/crowdplay/.

## 1 INTRODUCTION

Crowdsourcing has been instrumental in many AI advances, especially recent rapid progress in deep neural network models, which often rely on large training sets. For instance ImageNet (Deng et al., 2009), a large database of annotated images, has enabled a number of breakthroughs in image classification (Krizhevsky et al., 2012). At the same time, reinforcement learning (RL) has seen rapid progress in the last few years, fueled in part by the development of standard, easily accessible, benchmark environments like the Arcade Learning Environment (ALE) (Bellemare et al., 2013; Machado et al., 2018) and OpenAI Gym (Brockman et al., 2016). What has been underexplored is the intersection of the two: Using large-scale crowdsourced human data for offline learning, including imitation learning and offline RL.

We present *CrowdPlay*, a framework, methodology, and dataset that we hope will do for offline learning what ALE and OpenAI Gym did for online learning. CrowdPlay supports flexible and scalable crowdsourcing that is geared towards multi-channel recruitment, and is able to interface with any OpenAI Gym or Gym-like Markov decision process (MDP) environment. It supports real-time feedback to participants that can be used to boost data quality, as well as both purely human and mixed human-AI multiagent environments.

CrowdPlay is also the first dataset based on Atari 2600 games that features multimodal and multi-agent behavior. It includes both data from normal gameplay as well as explicitly multimodal behavioral data, where players are given instructions to follow a specific behavior. In addition to single-agent data, the dataset includes data from two-player, human-AI and human-human games, and with both competitive and cooperative rewards. Participants were recruited through multiple channels (under IRB, Harvard IRB18-0416) including Amazon Mechanical Turk, Lab in the Wild, undergraduate students, and multiple social media channels. For some platforms we also include data with a range of different incentive structures for participants. The Atari games were run using

ALE and a multiagent version of OpenAI Gym, guaranteeing that transitions are identical to what would be seen in standard Atari RL environments.

In this paper we focus on the use of CrowdPlay for Atari 2600 games, but a major advantage of the approach is that it works for any Gym-like MDP environment. We believe that Atari 2600 games are interesting for imitation learning (IL) and offline RL for the same reasons that they were seminal as a challenge problem in the development of RL: they offer a balance between achievable short-term research advances and sufficient complexity. More recent work in psychology has also shown them to be of sufficient richness to support the study of human learning behavior (Tsividis et al., 2017).

Further, and despite the richness of the data, it is easy to collect at scale through the use of crowd-sourcing and a web browser, and moreover, it can be used together with an established simulator for evaluation purposes. The CrowdPlay pipeline directly interfaces with standard RL environments. In particular this means that trajectories and transitions are guaranteed to be the same for human data as they are for RL environments; that offline learning methods automatically have access to a simulator; and that crowdsourcing with human players need not develop environments and tasks from scratch, but can make use of AI agents that can interact with human agents in a benchmark environment.

## 1.1 RELATED WORK

**Crowdsourcing**    Previous platforms for crowdsourcing such as *TurkServer* (Mao et al., 2012) have focused on supporting synchronous participation through Amazon Mechanical Turk participants and used to study economic behavior in simple experimental environments (and not for the generation of human behavioral data). Tylkin et al. (2021) use a combination of ALE and Javatari for crowd-sourcing Atari data in the context of evaluating the performance of an AI agent in human-AI collaboration. They propose modifying two-player Space Invaders to make it cooperative, and to train AI agents using randomized starting position, both of which we adopt in (some of) our multiagent environments. Their crowdsourcing approach is Atari-specific and not publicly available. Much work has been done on the study of incentives for participants in paid crowdsourcing studies, and this is also part of the focus of our work. Prior work (Mao et al., 2013; Mason & Watts, 2009; Yin et al., 2013; Harris, 2011; Shaw et al., 2011) has largely found that quality-dependent payments may increase quantity of work more than quality of work, and has not looked at real-time feedback on work quality.

**Offline Learning.**    Much work has been done on offline learning (Rashidinejad et al., 2021), including Behavior Cloning (BC) (Pomerleau, 1989), Batch Constrained Q-Learning (BCQ) (Fujimoto et al., 2019), Conservative Q-Learning (CQL) (Kumar et al., 2020), Implicit Quantile Network (IQN) (Dabney et al., 2018), DQN (Mnih, 2015) and Off-policy version of Soft Actor-Critic (Haarnoja et al., 2018). Aytar et al. (2018) demonstrate learning hard exploration games from unaligned human demonstration videos. Recent work (Schrittwieser et al., 2021) shows a sample-efficient model-based online and offline learning algorithm.

Specific to Atari, Kanervisto et al. (2020) benchmark behavioral cloning algorithms on existing data from several video games, including Atari 2600 games. Laurens & Kazmi (2021) clone River Raid agents using existing datasets, and develop evaluation metrics based on action distribution and playstyle.

**Datasets**    *Atari Grand Challenge* (AGC) (Kurin et al., 2017) is a dataset consisting of 45 hours of standard gameplay from five Atari 2600 games. The authors also make their Atari-specific data collection software available. They use a browser app running the Atari emulator in the browser based on Javatari. It is unclear to us if this can be guaranteed to always have identical execution to the ALE emulator.

*Atari-Head*  (Zhang et al., 2020) features 117 hours of gameplay data and includes eye-tracking data. This data was collected using ALE. However, the emulator was run in a semi-frame-by-frame mode, advancing the emulator state only when a key was pressed, and at a maximum of 20 frames per second. The focus of the study was on attention tracking, and is not intended to be representative of natural human gameplay behavior.

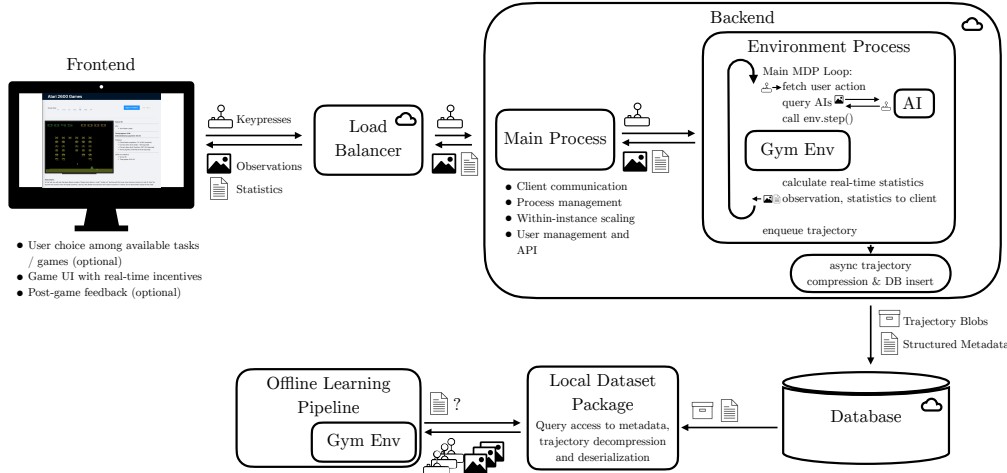

Figure 1: Key parts of the CrowdPlay software architecture. Arrows show the flow of keypresses, observations and metadata between browser client, MDP environment, AI policies, and the database, and the eventual flow to an offline learning pipeline. The backend, load balancer, and database are hosted on cloud infrastructure.

*D4RL* (Fu et al., 2020) and *RL Unplugged* (Gulcehre et al., 2020) both also provide datasets for offline learning, but both focus on synthetic data.

## 2 CROWDPLAY: THE PIPELINE

### 2.1 OVERVIEW

The heart of our pipeline are the CrowdPlay backend and frontend, which is a client-server architecture that streams OpenAI Gym environments and similar MDP environments to web browser clients. It is highly extensible, scalable to hundreds or thousands of concurrent users, and allows the real-time capture of both trajectories as well as related statistics. It is geared toward multi-channel recruitment of participants and strong incentives. As its most important feature, it interfaces directly with OpenAI Gym and similar environments, thus opening up the entire array of standard RL environments to rapid crowdsourcing of behavioral data into the support of research into IL and offline RL. It also supports multi-agent environments, including mixed human-AI environments.

Complementing this is an engine to support the local download of the generated dataset, including storage of metadata in a relational database for fast access, and compressed trajectories for storage efficiency. The data download can be real-time and is incremental.

We give a short overview of the CrowdPlay architecture, and refer the reader to Appendix A.1 for more information.

### 2.2 SOFTWARE ARCHITECTURE

CrowdPlay provides a highly extensible, high performance client-server architecture for streaming MDP environments to remote web browser clients. The backend interfaces with OpenAI Gym and similar MDP environments. Actions are collected as keypresses in the browser client, and sent to the backend where they are fed into the MDP's "step" function. The returned observation is sent back to the browser client for display. This is repeated to generate an episode trajectory. The remainder of the CrowdPlay software infrastructure is built to make this basic loop into a structure that is robust, performant, extensible, user-friendly and scalable. Figure 1 shows the key parts of the CrowdPlay architecture.

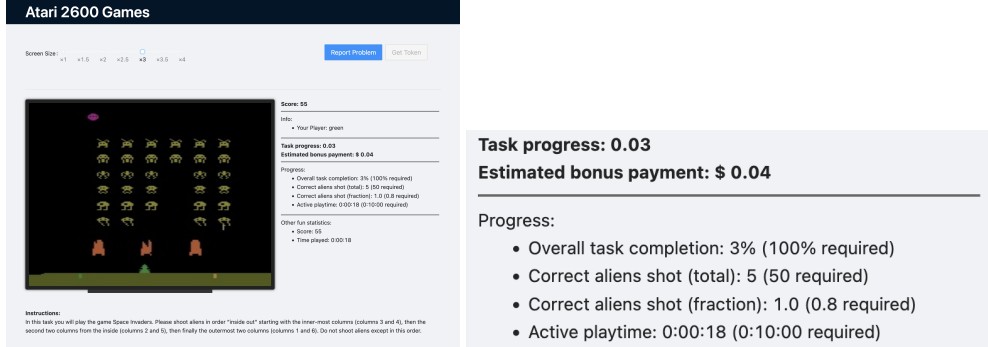

Figure 2: Screenshots of the main screen of CrowdPlay (left) and an enlarged detail of the realtime incentives (right).

Communication between the browser client and backend is through high-performance socket connections. The backend is built to be scalable both within-instance, using multiple processes, as well as across-instance using a load balancer and autoscaling instance groups. Trajectories are stored directly as compressed, serialized Python objects, allowing both very easy modification of data capture as well as immediate decoding for use in existing Python-based learning pipelines.

CrowdPlay also supports multi-agent environments. It allows multiple human participants by routing multiple browser clients to a single MDP environment. Mixed human-AI environments are supported through pre-trained neural network policies. For robustness, AI agents can also take over control of a human agent on the fly, in the case that a human player disconnects from a multiagent environment, allowing uninterrupted gameplay for the remaining human players.

A major focus in the design of CrowdPlay is providing the ease of access of the generated data in downstream ML pipelines. We believe it is crucial for these pipelines to have access not only to the same simulator as the crowdsourcing pipeline, but also to the same observation pre-processing tools that are used in state-of-the-art RL methods on these simulators. Addressing these design goals, CrowdPlay includes a local metadata search engine and a custom, Deepmind-style (Mnih et al., 2015) observation processing function for offline data. We give more details in Appendix A.1.

CrowdPlay provides an extensible and easy-to-use framework for collecting structured metadata and real-time statistics per user, session, episode, and individual steps. This is used for capturing data quality information, as well as for driving real-time incentives for participants, depending on recruitment platform. We discuss the platforms that we target in more detail in Appendix A.3, and the various incentive designs and their effect on data in Section 4.

## 3 CROWDPLAY ATARI: THE DATASET

### 3.1 SCOPE AND MOTIVATION

Our main focus in creating the first CrowdPlay dataset has been on Atari 2600 games, as we believe that human data for these environments can enable advances in IL and offline RL (just as they have been in driving advances in online RL, and for the same reasons). In curating the dataset, we have been especially motivated by diversity—we have used multiple recruitment channels, each with different extrinsic and intrinsic user motivation and incentive models, and have reached over 800 users in 1300 sessions over a three week period in September 2021. The CrowdPlay Atari dataset currently holds over 250 hours, or 54 million transitions, and was collected across six different games. We have targeted not only implicitly multimodal behavior through recruiting different users and through different channels, but also explicit multimodal behavior through explicit instructions that are reinforced through tight incentives and real-time feedback. We include both single agent as well as multi-agent data, with the latter reported for both human-human and human-AI gameplay, and with both competitive as well as cooperative rewards. We believe this is the first multi-agent human behavioral dataset, at the very least for Atari 2600 games.

## 3.2 Task Design

We used the following Atari 2600 games for our dataset: Space Invaders, River Raid, Montezuma's Revenge, Q*bert, Breakout and Beamrider. Space Invaders makes up the largest part of our dataset. We chose this game as a focus for several reasons: it is well-known and popular both among Atari players as well as in the RL community, it was easy to come up with several specific and distinct behaviors that are still compatible with progressing in the game, and it has a native two-player mode. River Raid provides the second largest amount of data and was chosen for similar reasons, in that it is accessible and well understood from a learning perspective, and has obvious opportunities to promote multimodal behavior. The remaining games were chosen for diversity of game types and their popularity in previous RL and IL work. Table 1 shows a list of available data by game and recruitment channel.

**Multimodal Behavior**   For Space Invaders and River Raid, in addition to standard gameplay, we asked some participants to follow specific behavior in the game. We believe that this data will be useful for multiple research strands. In imitation learning, this type of data can provide a testbed to understand whether algorithms are robust to divergent expert behavior. In offline RL, this can allow for controlled experiments with different reward functions in an otherwise identical environment. For most of this data, we recruited participants via Mechanical Turk, and gave incentives via both a minimum level of task adherence required to complete the HIT (unit of task), as well as an explicit reward function tied to a bonus payment, and with real-time feedback on the same. The behavior types we instructed participants to follow were to either stay on either half of the game screen (in both Space Invaders and River Raid, or to shoot aliens in a particular order (row by row from the bottom, column by column from the outside in, or column by column from the inside out; in Space Invaders only.) We discuss these tasks in more detail in Appendix A.2.1

**Multiplayer Games**   CrowdPlay also contains a set of trajectories from multi-agent environments. For these, we used the two-player mode of Space Invaders. We include data from two variants: (1) the standard two-player Space Invaders mode, and (2) a custom cooperative mode. In the latter, we modify the original game in two ways. First, in the standard two-player mode there is a score bonus if the other player loses a live. We removed this score bonus. Second, we gave both players the sum of their individual scores. These modifications were done entirely in Python, in the case of the score bonus detecting these events from emulator state. Participants were instructed to ignore the score shown on the game screen, and were instead shown their "cooperative" score next to the game screen. Most of our multiplayer data is from mTurk, and people were incentivized through bonus payments to maximize this score. Further, we include data from games with two human players, as well as games with one human and one AI player. We give details on this in Appendix A.2.6

## 3.3 Data Analysis

A unique feature of the CrowdPlay Atari dataset is its size and diversity. In addition to explicit multimodal behavior, it also comprises data generated by participants recruited via multiple different channels, and different demographics. Figure 3 (left) shows a t-SNE embedding of action distributions in episodes of standard Space Invaders gameplay. Episodes are colored according to recruitment channel (MTurk, Email or social media). We notice that there is a clear divide between action distributions of MTurk users and those of Email and social media users, which we take as evidence that different demographics already lead to interesting diversity in the data.[1] This shows that different recruitment channels lead to diversity in the data, this already apparent in aggregate statistics such as action distributions.

For multiagent data, Figure 3 (right) shows a t-SNE embedding for cooperative and standard human-AI Space Invaders for MTurk users. We note several things. First, there is a clear distinction between human and AI actions. Second, there are two clear clusters within the AI data that correspond to cooperative and competitive behavior. Third, there is a clear relationship between cooperative and

---

[1]The mostly-orange (email users) cluster on the left corresponds to episodes where no key was pressed at all, while the green (MTurk) cluster at the bottom right corresponds to episodes where the fire button was held the entire episode (and no other key was pressed). The minimum requirement to complete the task for each group had the same requirement of ten minutes of active playtime, which the no-keypress episodes in the orange cluster would not contribute toward.

Table 1: Dataset by game and recruitment channel

| Task | Data Collected (hours) | | | |
|---|---|---|---|---|
| | MTurk | Social Media | Email Raffle | Total |
| **Beamrider** | **7.90** | **-** | **-** | **7.90** |
| **Breakout** | **11.45** | **-** | **-** | **11.45** |
| **Montezuma's Revenge** | **16.70** | **3.75** | **5.19** | **25.65** |
| **Q*Bert** | **6.97** | **2.90** | **-** | **9.87** |
| **Riverraid** | **17.64** | **4.47** | **3.10** | **25.20** |
| - of which multimodal | 12.29 | 0.69 | 1.10 | 14.08 |
| **Space Invaders** | **196.09** | **18.34** | **7.10** | **221.53** |
| - of which incentives | 81.41 | - | - | 81.41 |
| - of which multimodal | 101.46 | 0.52 | 1.12 | 103.09 |
| **Space Invaders (2P)** | **6.35** | **0.38** | **0.81** | **7.54** |
| **Space Invaders (2P w/AI)** | **54.98** | **2.54** | **1.41** | **58.93** |
| **Total** | **318.07** | **32.38** | **17.60** | **368.05** |

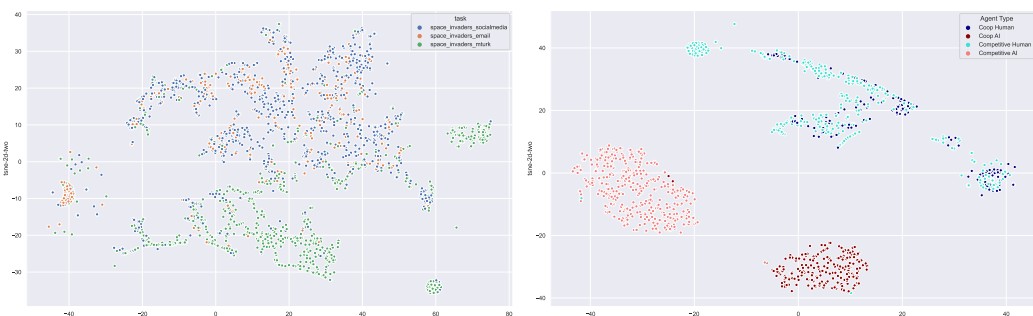

Figure 3: t-SNE embedding of action distributions of different (human) participant types in single-agent games (left), and human and AI agents in cooperative and standard multiagent games (right).

standard settings and action distributions for human data and this is more complex than in the AI data.

### 3.4 CHALLENGES AND LIMITATIONS

We discuss here some challenges that we experienced in collecting this data and the corresponding limitations in the dataset. One, our incentive design has continued to evolve, while a significant amount of the data has been collected with suboptimal incentives. This is part of why we discuss incentive design in the next section in detail, to allow other researchers to build on what we have learned. Two, some of our dataset is unbalanced, especially for the different behavior types in the multimodal Space Invaders data, where we have 2-3 times as much data for some behaviors as for others. This is partly due to suboptimal incentives and some tasks being harder than we had intended, and partly because we improved the way the pipeline assigned MTurk workers to tasks over time. We have since then improved the automatic task assignment logic to more carefully take into account the amount of data already collected. Three, we have found that collecting human-human multiagent data is a difficult challenge; e.g., we expected that the ability to play with friends would be a major draw in social media recruitment, but found that we had virtually zero uptake on this. On the other hand, we have made significant advances in making multiplayer data collection more feasible on MTurk. In particular, the use of fallback AI agents has solved many practical problems with workers needing to depend on one another for successful task completion. Still, we were able to collect much more single-player and human-AI data than human-human data.

## 4 INCENTIVE DESIGN

### 4.1 INCENTIVE DESIGN AND RECRUITMENT

CrowdPlay has been designed with multichannel recruitment and strong participant incentives in mind. It supports configuring multiple environments and "tasks" for participants to complete, including different user instructions, incentives, and AI agent configuration. At the heart of its capabilities is its ability to capture an extensible list of metadata live during gameplay, including data such as playtime, score, and various in-game behavioral characteristics. On platforms where users are paid for their participation, we use this data to define both a minimum acceptable effort by users to get paid at all, as well as a dynamic bonus payment that can depend on a fine-grained analysis of user effort. Both progress toward minimum required effort as well as bonus payments can be displayed to users during live gameplay. On platforms where users are not compensated, we use the same real-time statistics to reinforce intrinsic motivation through live "high score" pages and by reframing non-standard gameplay tasks as "challenges."

For instance, in one Space Invaders task we asked participants to shoot aliens in a specific order. Our architecture is able to determine adherence to these instructions by evaluating emulator state at every frame, detecting when an alien has been shot and which one, and keeping a running count of aliens shot in and out of the prescribed order. Users were rewarded based on both the total number of aliens they shot in the correct order, as well as the fraction of aliens that they shot correctly. Using this framework, we can mimic the reward structure of an MDP through monetary incentives and also make use of realtime feedback; and we can additionally shape incentives to achieve particular modalities. Figure 2 (right) shows an example of this realtime feedback.

### 4.2 INCENTIVE MODELS

We provide various incentive models, as well as real-time feedback to participants. For social media participants, this was feedback-only. For students we made use of a raffle, and required a minimum time and effort, e.g. 80% of aliens shot in the correct order. For participants from Mechanical Turk, we experimented with multiple models of incentives.

**Experimental design** Specifically, we report the results of an extensive experiment with the Space Invaders "Inside-Out" task. In this, we gave participants identical instructions for the behavior they were asked to follow, but adopted five different ways of remunerating them: (1) No incentives (payment for everyone), (2) Active Time (payment subject to non-trivial behavior), (3) Minimum Requirement (payment subject to minimum performance requirement), (4) Sliding Bonus (variable payment depending on performance) and (5) All Incentives (sliding bonus contingent on meeting minimum requirement). Appendix A.4 details these.

For both task requirements and bonus payments, participants were given real-time feedback. For minimum requirements, this took the form of an itemized list of requirements and their performance, e.g. "Correct aliens shot: 40 (required: 50)." For bonus payments, this took the form of a live estimate of the expected payment, but did not include details on how the payment was computed. Task instructions stated that the bonus payment would depend on both the number of aliens shot as well as how many were shot in the correct order.

We also compare data from these treatments with data collected from social media users, as well as students reached on campus via email. For email participants, we enforced a similar minimum requirement as for the minimum requirement treatment on MTurk, with successful completion of the requirement earning an entry into a raffle. For social media participants, we did not provide any monetary incentives, and instead phrased the task as a "challenge."

### 4.3 RESULTS

In regard to the structure of incentives and effect on data, we report several main findings. First, among paid participants, data quality (measured as the fraction of data per participant meeting a threshold of at least 80% of aliens shot in the specified order) was significantly higher when using any kind of quality-based incentives, be it a minimum requirement tied to a fixed-amount payment ("quality requirement"), a sliding bonus payment ("sliding bonus"), or a combination of both ($p <$

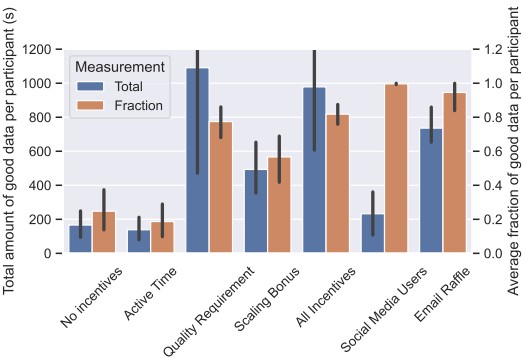

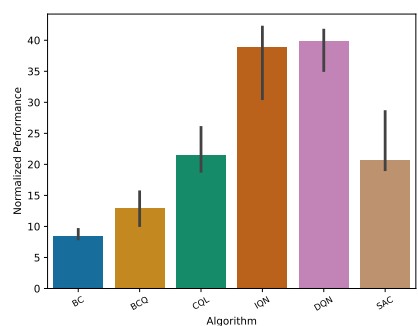

Figure 4: Data quality for different incentive treatments and recruitment channels. Blue bars show the total amount (in seconds) of "good data" collected per user, where this is defined as episodes with at least 80% task adherence. Orange bars show the fraction of good data compared to the total data collected per user.

Figure 5: Evaluation Performance of offline RL algorithms across different tasks. The bars indicate the median normalized score across tasks, and the error bars show a bootstrapped estimate of the $[25, 75]$ percentile interval for the median estimate. The score normalization is computed using the best score achieved by humans across each task.

.05 for any pair of treatments, comparing with either of "no incentives" or "active time"). See Figure 4 (orange bars). Non-incentivized data was low quality even when enforcing that participants actively play the game, thus preventing users from putting in no effort at all (e.g., by starting the game and then switching to a different window).

Second, participants recruited via social media had the highest data quality (statistically significant at $p < .05$ for pairwise comparison with any other treatment), but we also found this channel to be the least scalable (see Appendix A.2.4). Users recruited via an email raffle had nearly as high data quality, but here the difference against the quality-based incentive treatments is insignificant.

Third, among paid participants the incentive treatments led to a larger amount of good data being collected per participant ($p < .05$ for any pair of treatments, comparing with either of "no incentives" or "active time"). See Figure 4 (blue bars). The "quality requirement" and "all incentives" treatments showed the highest amount of good data per participant, although the comparison with "sliding bonus" is not significant (due to some outliers in both treatments).

## 5 BENCHMARK RESULTS

Our framework provides a variety of data compositions that exhibit real-world intricacies and diversities. We envision that this will help facilitate rapid progress for offline learning methods that rely on learning from diverse data compositions without access to online exploration (Rashidinejad et al., 2021). We therefore evaluate our datasets on recently proposed offline learning algorithms and hope that it will inspire potential future directions in this area. As Atari is a discrete action domain, we focus on algorithms that can handle discrete actions effectively. Specifically, our baseline algorithms include Behavior Cloning (BC) (Pomerleau, 1989), Batch Constrained Q-Learning (BCQ) (Fujimoto et al., 2019), Conservative Q-Learning (CQL) (Kumar et al., 2020), Implicit Quantile Network (IQN) (Dabney et al., 2018), DQN (Mnih, 2015) and an offline version of Soft Actor-Critic (Haarnoja et al., 2018). As we evaluate the algorithms on different tasks, in Figure 5, we provide the normalized median performance across tasks (Agarwal et al., 2020) where the normalization is done using the best performance achieve by humans on corresponding games.

From the results, we observe that the recent advancements in offline reinforcement learning algorithms contribute towards outperforming behavioral regularized algorithms. Further, we found that the off-policy algorithm DQN serves as a strong baseline for offline RL albeit with a higher variance across tasks and seeds. Overall, the results for all the algorithms demonstrate the open challenges on learning from human demonstrations where the data is both limited and noisy which raises the need for more robust and sample efficient methods. First, we contrast our results with benchmark

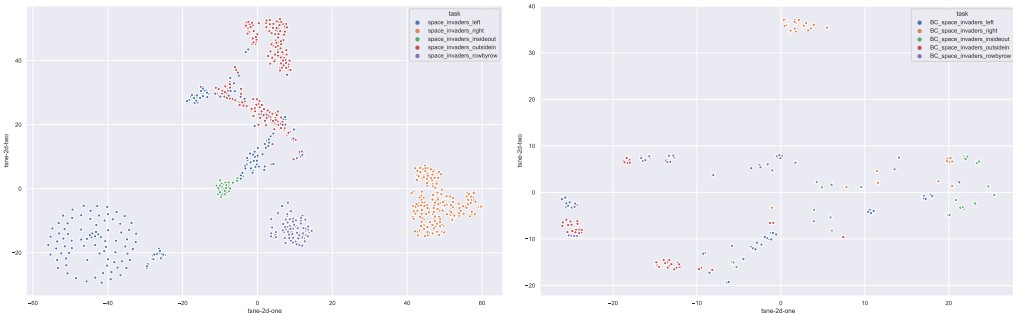

Figure 6: Separate t-SNE embeddings of action distributions and behavioral statistics in multimodal behaviors for human participants (left) and BC agents (right).

results in the RL Unplugged paper (Gulcehre et al., 2020), where a subset of these algorithms were evaluated on specific games we benchmark here but the demonstrations were collected using an online DQN agent trained to maximize the game performance. Their results show that both in general (Figure 6 therein) and on the specific games we benchmark here (Table 9 and Figure 7 therein), the offline algorithms performed significantly and consistently better than the original policy performance. In comparison, none of the algorithms in our experiments were able to achieve better than around $45\%$ of best human performance on average (with the exception of IQN on couple of tasks), and all algorithms demonstrated high variance in performance across seeds and tasks. Second, conservative algorithms that constrain the policy to the dataset (e.g. BCQ) and are tailored to address erroneous optimistic value function estimation in the presence of complex and multi-modal distributions (e.g. CQL) do not perform well on data collected from human demonstrations. This is in sharp contrast to their high performance reported on datasets collected using AI agents, as reported in Fu et al. (2020). This demonstrates that human data poses an interesting and distinct challenge to state-of-the-art offline learning algorithms compared with synthetic data.

Details on the experimental setup, performance of algorithms on a suite of robust metrics, individual normalized and unnormalized score for each task and algorithm, and task-specific performance profile for each algorithm are provided in Appendix A.7.

Beyond performance comparisons, offline algorithms also struggle to qualitatively capture human data. Figure 6 shows separate t-SNE embeddings of behavior of human participants (left) and AI agents trained using BC (right) for the five explicit multimodal behavior types in Space Invaders. We see that while human data is clustered in clearly distinguishable regions for the behavior types, this is not the case for the BC agents. This shows that multimodal human data poses a challenge for offline learning algorithms that aim to emulate human behavioral characteristics such as BC and other imitation learning methods. See Appendix A.6 for details on t-SNE parameters and data.

## 6 DISCUSSION

We have presented CrowdPlay, a pipeline for crowdsourcing human demonstration data in standard RL environments, and a first dataset together with benchmarks of offline RL algorithms on Atari 2600 games. A key advantage of CrowdPlay is that it allows easily accessible crowdsourcing for any existing RL environment, and it can provide realtime feedback and incentives that mirror arbitrary RL reward functions or other kinds of desired, complex behavior types. Further, by using standard RL environments we can streamline the end-to-end crowdsourcing and learning pipeline: CrowdPlay can use the same environment simulator that is used by an existing RL pipeline, enabling for instance mixed human-AI environments that use the same algorithms already used to train AI agents. We hope that this new framework and dataset will enable rapid progress in offline learning on large-scale human demonstrations. Atari 2600 games were seminal in driving progress in online RL, and we believe this dataset and pipeline can advance IL and offline RL research. Moreover, this pipeline is not constrained to Atari games, and we hope that it will be utilized for other environments. Beyond supporting progress in IL and offline RL, we also believe that CrowdPlay can enable other novel applications, both in computer science (such as task classification), but also in other fields such as psychology or behavioral economics.

ETHICS STATEMENT

An ethical concern in regard to crowdsourcing is about fair compensation for participants.

The real-time feedback to participants on our platform is designed to help with this. It provides fine-grained information on what is required to be compensated (Figure 2 right, for example) including real-time information on progress and on the payments participants will receive. This removes information asymmetry, helping to avoid submissions that appear low-effort due to misunderstanding, and also supporting informed decisions about continued participation. Further, it creates accountability: we commit to a compensation scheme that is visible to participants and promise payments that are communicated in real time.

With one exception, where a task was harder than we expected and we later lowered the requirements, we did not receive significant negative feedback from mTurkers regarding the task. Of those that submitted a HIT, the feedback was overwhelmingly positive and often described the task as "fun" or similar. Still, we want to highlight that care should be taken in designing compensation requirements. An unintentional mistake we made in the "minimum requirement" treatment in the incentives experiment was to not provide a base payment for participation, coupling this with what we thought was a very low task requirement that would only act to filter out spam. We later noticed a number of workers dropping the task, and leaving without payment. We do not know for sure why this is — it is possible workers got bored, but it is also possible that some workers struggled to meet the minimal requirements, despite our intention of setting a very low bar. We did not include data from these users in the published dataset. For future studies we recommend care in deciding on minimum requirements, and the use of a small base payment that rewards active playtime. In a small number of cases, a worker contacted us about a task they were unable to complete (for instance due to technical issues or difficulty). We always opted to compensate workers in such cases.

The above is largely relevant for recruitment channels where there is some form of compensation (e.g., monetary compensation such as on MTurk, or a raffle among participants). For the specific Atari dataset we collected, because these videogames are inherently "fun", we also worked with unpaid participants recruited from social media. We took care to target potential participants who were already looking for videogames to play online (for instance, by posting in a community discussing "web games" on the social media site Reddit). Further, we took care in selecting the specific experiments offered to these participants. Specifically we offered the standard gameplay variants, with no minimum time limits or any other restrictions. We offered one of the multimodal behaviors as a "challenge". We specifically chose the "inside-out" behavior type in Space Invaders for this experiment: while doing so is challenging, it is still possible to progress in the game while following the behavioral instructions. We also did not enforce any adherence to this behavior type. Behavior types that are incompatible with game progress might be frustrating for users, and we consider those inappropriate for unpaid participants. Similarly for domains other than videogames, researchers should carefully consider whether an experiment is appropriate for unpaid participants.

Our data collection underwent IRB review at our institution (Harvard IRB18-0416), which included recruitment and compensation details.

REPRODUCIBILITY STATEMENT

Our entire framework and current version of the dataset are available publicly at `https://mgerstgrasser.github.io/crowdplay/` under an open-source license and we have provided the links to the implementations used to produce the benchmark results in Appendix A.7.

ACKNOWLEDGEMENTS

We thank Francisco Ramos for his help in implementing the CrowdPlay software, especially on the frontend. We would like to thank anonymous reviewers at ICLR 2022 for their constructive feedback and discussions. This research is funded in part by Defense Advanced Research Projects Agency under Cooperative Agreement HR00111920029. The content of the information does not necessarily reflect the position or the policy of the Government, and no official endorsement should be inferred. This is approved for public release; distribution is unlimited.

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

# A  APPENDIX

## A.1  TECHNICAL DETAILS

**Architecture Overview**    CrowdPlay uses a client-server architecture, where an MDP environment is run in a simulator and its output streamed to a web browser client. The browser client collects user keypresses which are streamed to the server, where they are decoded into actions which are fed into the MDP simulator. The backend hosts the MDP simulator, records trajectories into a database, and provides additional functionality. Communication uses a http API for management tasks such as establishing a connection, and a websockets for streaming keypresses and observations for better performance. The backend can optionally interface with an external load balancer and MySQL instance for scalability.

**Backend Architecture**    The backend is written in Python for best compatibility with existing Python-based RL simulators and easy data interchange with offline learning pipelines. It is highly parallelized across multiple processes: A main process handles http requests to the API, routes websockets communications, and manages the multiple concurrent MDP environments. Each environment simulator is run in its own process dedicated in essence to just the main "fetch action - environment step - get observation" MDP loop. Once an episode is complete, the entire trajectory is asynchronously serialized using Pickle and compressed using Bzip2 in a separate process and inserted into a database.

**MDP simulator interface**    CrowdPlay interfaces with standard MDP simulators such as OpenAI Gym environments, but also similar interfaces. For our current set of experiments, we use a Gym-like interface implemented using RLLib's multiagent environment definition, and built on a multiagent version of ALE Terry et al. (2020). We use this even in single-agent games, although this is for convenience and not due to an architectural limitation. The main requirements for the MDP simulator are that it provides a reset () and a step ( action ) function.

**Frontend**    The CrowdPlay frontend is written in React for a modern UI experience. Its main task is to receive image observations and associated metadata (such as real-time task progress, estimated bonus payment and other statistics) and display them to the user. Optionally users can choose the task / game they wish to play in a library-style user interface; users can be shown a post-game "high score" page which shows performance and other game statistics relative to other participants; and users can report technical problems and leave feedback. For multiplayer games, the frontend supports waiting rooms and can notify players who joined early once enough other players have joined to start the game.

**Environment and task definitions**    CrowdPlay supports multiple environments and environment variations concurrently. Each variation or "task" can include a different base environment (e.g. a different Atari game, or a different MDP simulator altogether), different instructions for the user, different incentives and statistics (see below), different configurations of human and AI players in multiplayer games (see below) etc. For instance, one task could be a two-player Space Invaders game with one human player and one AI agent, with the user instructed to follow a particular behavior in game, and a minimum effort requirement and bonus payment tied to their adherence to that behavior (for deployment to paid participants); while another task could be a two-player Space Invaders game with two human players and no incentives (for deployment to unpaid participants). Users can be assigned to a task through a URL parameter. Tasks can also be grouped into task groups. Users can then either select a task of their choice among a task group; or the backend can automatically assign a task. This automatic assignment can take into account the amount of high-quality data already collected for each task, and can either aim to collected a minimum amount for each task before assigning incoming users to the next task, or to collect an equal amount of data for each task. Quality is determined by episode and session-level realtime statistics (see below). If any tasks in a task group are multiplayer tasks, the scheduler will first try to assign users to any already-instantiated environments that are waiting for additional human players, before considering creating a new environment.

**AI agents**   Because CrowdPlay uses standard MDP interfaces also used in established RL methods, it can support AI agents trained with those methods. We currently have an implementation for policies trained with RLLib, which would be easy to extend to other existing libraries like OpenAI Baselines. CrowdPlay includes a custom image preprocessor that replicates the behavior of standard "DeepMind" wrappers such as those included in OpenAI Gym, but without having to wrap around the MDP environment. This allows us to provide AI agents with a standard "four frames stacked, each four frames apart" observation, while providing human players with an unprocessed observation every frame.

**Multi-player**   Multi-agent environments are supported with both multiple human players as well as mixed human-AI games. For human players, the frontend routes multiple users' websocket connections to the same environment process. AI agents are implemented as above. As a key robustness feature, AI agent can take over on-the-fly if a human player disconnects, which enables the remaining human players to continue playing. This is important on platforms such as Amazon Mechanical Turk, so that one participant disconnecting does not leave another participant unable to complete their task through no fault of their own. In such environments, we create fallback AI policies for all agents when the environment is started, to avoid an interruption during gameplay (instantiating the policy takes a few seconds). However we only start observation preprocessing and querying the AI policy for actions for each agent only if and once the human player controlling that player disconnects.

**Real-time statistics engine**   CrowdPlay can calculate and record gameplay statistics and other metadata using an extensible real-time statistics engine. The information to be recorded can be configured for each task. CrowdPlay ships with several generic statistics collectors such as for score and active playtime, and several game-specific collectors such as the player's x-position in Space Invaders and River Raid, and the order in which aliens have been hit in Space Invaders. These can be used to determine how closely the player's gameplay behavior matches a given target behavior. In turn, CrowdPlay supports minimum effort requirements to consider a task "complete" as well as sliding bonus payments based on the statistics. For each statistic defined for a task, per-step information is saved as part of the trajectory in a compressed binary blob, while episode-level and session-level aggregate statistics are saved as structured metadata in a relational database to allow for fast filtering of individual trajectories in downstream offline learning pipelines. Statistic collectors are implemented as functions or Python callables that take as input the trajectory information at each step, including actions, observations and emulator state. Because all of this information is saved, statistic collectors can also be run offline against already recorded trajectories should requirements change later.

**Deployment and Scalability**   The CrowdPlay backend is fully containerized using Docker and can be deployed on any compatible platform. Within-instance scaling to multiple processor cores is handled within the backend. CrowdPlay also includes configuration options for deployment using Amazon Elastic Beanstalk (EB). When deploying on EB, a load-balancer can be used, which enables auto-scaling across instances (additional server instances will be started automatically depending on load). This enables CrowdPlay to scale to hundreds or thousands of simultaneous users. EB can also be configured to use AWS Spot Instances for significantly reduced operating cost.

**Latency**   CrowdPlay records at every keypress event the sequence number of the frame shown on screen at the time of the keypress. This allows us to measure end-to-end latency of the system as the difference between the current frame sequence number in the back end when the keypress is received there, and the sequence number of the frame shown to the user when the key was pressed. Across our current Atari dataset, we measured a median latency of 7 frames or 116ms, and downstream learning pipelines could filter for acceptable levels of latency should this be a concern.

**Dataset Engine**   Recorded trajectories and metadata can be downloaded into a locally installable Python package providing user-friendly metadata search and trajectory loading functionality. Download from the backend MySQL database can be online and incremental. Metadata is saved locally in a SQLite database, and is mapped into Python classes by the dataset package. This allows filtering episodes by metadata using either SQL queries, or native Python techniques (e.g. list comprehension) according on user preference. Trajectories can be stored in the bzip2-compressed format used

by the backend, or expanded into uncompressed pickle files for much faster loading at the expense of storage efficiency.

**Observation Processing**   Most Atari RL pipelines use some variation of "Deepmind"-style observation preprocessing as pioneered in Mnih et al. (2015). This includes taking the pixel-wise maximum of two consecutive frames, skipping several frames between observation, rescaling images to 84x84 grayscale, and stacking four such consecutive observations. Frameskipping in particular is a challenge in our context as it effectively reduces the time-resolution of the environment by a factor of four (one observation given for every step in the Atari emulator), which we do not want to do for human players, but still need to do for AI players in mixed human-AI multiplayer games. Similarly, all these processing steps are usually implemented as environment wrappers, which aren't directly applicable to offline data. We therefore include a custom implementation of the above. This is implemented through a custom frame buffer that replaces the usual "Deepmind" wrappers. Our implementation keeps a buffer of the latest several raw observations and can generate a max-and-skipped, rescaled, framestacked observation on the fly. The implementation is verified to give the same output frame-by-frame and pixel-by-pixel as standard implementations such as those in OpenAI Gym, and can be applied post-hoc to already recorded trajectories. This same observation processing is also used for the AI agents that we bring into mixed human-AI environments.

## A.2   DATASET DETAILS

### A.2.1   GAME VARIANTS

We instructed participants to follow these five behavior types:

**Left-side-only**   In both Space Invaders and River Raid, we instructed participants to score as high as possible in the game, while staying only on the left half of the game screen.

**Right-side-only**   As above, but staying on the right side of the game screen.

**Row-by-row**   In Space Invaders we instructed participants to maximize game score while aiming to shoot aliens row by row from the bottom; i.e., participants were asked to completely clear the bottom-most row of aliens, then the second-from-the-bottom row, etc.

**Outside-in**   In Space Invaders, we instructed participants to maximize game score while aiming to shoot aliens column by column from the outside in; i.e., participants should completely clear the two outermost columns, before moving on to the next two columns, etc

**Inside-out**   In Space Invaders, we asked participants to shoot aliens column by column, from the inside out; i.e., participants should completely clear the center two columns (column 3 and 4), then the next two columns from the center (columns 2 and 5), and finally the two outermost columns (1 and 6).

### A.2.2   FULL LIST OF INCLUDED DATA

Table 2 shows a detailed list of available data by game, game variant, and recruitment channel.

### A.2.3   DATA EVALUATION

The dataset contains rich metadata accessible through both SQL queries as well as Python classes. Metadata includes universal attributes like total episode score and length, but also task-specific attributes like total and fraction of aliens shot in the correct order in the above Space Invaders tasks. This allows fine-grained filtering of available trajectories according to specified criteria. We propose that "data quality" should be thought of as multidimensional. For instance, we saw in our multimodal task data both trajectories with moderately high task adherence (as fraction of aliens shot in correct order) and very high overall performance and length, and vice versa trajectories of moderately high performance and very high task adherence. Our dataset allows filtering for minimum levels of both attributes, as well as others. Further, any metadata not saved explicitly as such could be recovered from the trajectories themselves, as the full emulator RAM is stored in addition to observations.

Table 2: List of all tasks in the dataset

| Task | Data Collected (hours) | | | |
|---|---|---|---|---|
| | MTurk | Social Media | Email Raffle | Total |
| **Beamrider** | **7.90** | **-** | **-** | **7.90** |
| **Breakout** | **11.45** | **-** | **-** | **11.45** |
| **Montezuma's Revenge** | **16.70** | **3.75** | **5.19** | **25.65** |
| **Q*Bert** | **6.97** | **2.90** | **-** | **9.87** |
| **Riverraid** | **17.64** | **4.47** | **3.10** | **25.20** |
| - plain | 5.35 | 3.78 | 2.00 | 11.12 |
| - left | 6.78 | 0.45 | 0.70 | 7.94 |
| - right | 5.51 | 0.23 | 0.40 | 6.14 |
| **Space Invaders** | **196.09** | **18.34** | **7.10** | **221.53** |
| - plain | 13.22 | 17.83 | 5.98 | 37.03 |
| - left | 18.80 | - | - | 18.80 |
| - right | 36.35 | - | - | 36.35 |
| - insideout | 16.49 | 0.42 | 1.12 | 18.03 |
| - outsidein | 13.44 | - | - | 13.44 |
| - rowbyrow | 16.38 | 0.09 | - | 16.47 |
| - incentives (insideout) | 81.41 | - | - | 81.41 |
| **Space Invaders (2P)** | **6.35** | **0.38** | **0.81** | **7.54** |
| - competitive | 2.60 | - | 0.42 | 3.02 |
| - cooperative | 3.75 | 0.38 | 0.39 | 4.53 |
| **Space Invaders (2P w/AI)** | **54.98** | **2.54** | **1.41** | **58.93** |
| - competitive | 45.10 | 1.80 | 1.06 | 47.95 |
| - cooperative | 9.88 | 0.74 | 0.35 | 10.97 |
| **Total** | **318.07** | **32.38** | **17.60** | **368.05** |

### A.2.4 RECRUITMENT

We recruited participants through Amazon Mechanical Turk; emails to undergraduate students; and several social media platforms. Participants recruited via MTurk were paid a variable payment depending on their performance. Participants recruited via email were entered into a raffle for Amazon gift cards for each task they successfully completed. Social media users were not compensated for their time.

Recruitment on MTurk was the most scalable (in terms of ease of collecting large amounts of data), and recruitment via social media the least. For social media, the amount of data collected per participant was the lowest of all channels. The total number of users from social media and email was also lower than those from MTurk, and it would easily be possible to recruit more MTurk users. Furthermore, a large fraction of our social media data came from a single post on Reddit, and even with this relatively few social media recruits opted to pursue tasks other than standard gameplay.

### A.2.5 EXAMPLE PARTICIPANT INSTRUCTIONS

**Standard Gameplay** For standard gameplay, we provided participants with instructions on how to play the game, technical information and platform-dependent information on payment. For instance, for Space Invaders, our instructions to participants recruited via social media read as follows:

> In this experiment you will play the game Space Invaders. Your goal is to shoot as many of the alien invaders shown on the screen as possible. You control the small cannon at the bottom of the screen. Use your keyboard's arrow keys to move your cannon left and right, and use the spacebar key to fire. The aliens will

fire at you too. If you are hit by alien fire three times, the game ends. The game will also end if the alien invaders reach the bottom of the screen. A new game will then start automatically, but you can end the experiment clicking the Finish Experiment button on the top right. You can end the experiment at any time, but we encourage you to play at least a few games.

For participants recruited via MTurk, the instructions read:

> In this experiment you will play the game Space Invaders. Your goal is to shoot as many of the alien invaders shown on the screen as possible. You control the small cannon at the bottom of the screen. Use your keyboard's arrow keys to move your cannon left and right, and use the spacebar key to fire. The aliens will fire at you too. If you are hit by alien fire three times, the game ends. The game will also end if the alien invaders reach the bottom of the screen. A new game will then start automatically. You must play for at least 10 minutes and show some effort in the form of a minimum score to complete this task. Your bonus payment will depend entirely on your score.

**Multimodal Behavior**  For specific multimodal behavior we showed specific instructions:

> In this task you will play the game Space Invaders. Please shoot aliens in order "inside out" starting with the inner-most columns (columns 3 and 4), then the second two columns from the inside (columns 2 and 5), then finally the outermost two columns (columns 1 and 6). Do not shoot aliens except in this order.
>
> User your keyboard's arrow keys and space bar to interact with the game. It is up to you to follow the instructions above, please follow them carefully.

**Multiplayer Games**  For multiplayer games, we provided these instructions:

> In this task you will play the game Space Invaders. This game is cooperative - you and the other player both get points if either of you hits an alien. Please IGNORE THE SCORE SHOWN ON THE GAME SCREEN, and try to maximise the score shown in the text box to the right of the game screen. This score is shared between you and the other player. You may want to cooperate with the other player.

In all cases, specific instructions were followed with these generic instructions:

> User your keyboard's arrow keys and space bar to interact with the game. It is up to you to follow the instructions above, please follow them carefully.
>
> Please do not open multiple tabs or windows of this link at the same time. In a two-player task, if you are not being connected to another human player, or there appear to be issues (e.g. other player is not moving at all), please click 'Reload' in your browser window and tell the other player to do the same.

Instructions for other games and tasks were similar.

### A.2.6  MULTIAGENT DATA

**Game Variants**  The CrowdPlay Atari dataset contains data from both standard two-player Space Invaders as well as a custom cooperative mode. These differ in two ways: Firstly, in the standard variant there is a bonus score when the other player gets hit by alien fire. We remove this score bonus in the cooperative variant, in order to remove incentives to hurt the other player. Secondly, we give both players the average of their clipped rewards instead of the usual rewards. For instance if one player hits an alien and gets an unclipped reward of 15, while the other player does not get a reward in the frame, we first clip the rewards (from 15 and 0 to 1 and 0, respectively), and then average the clipped rewards to give each agent a reward of 0.5. In the standard non-cooperative variant, the players would receive rewards of 1 and 0, respectively.

**Fallback AI Agents**  AI agents were also used in nominally human-human environments with MTurk participants, in two circumstances. First, if one human player disconnected, an AI agent

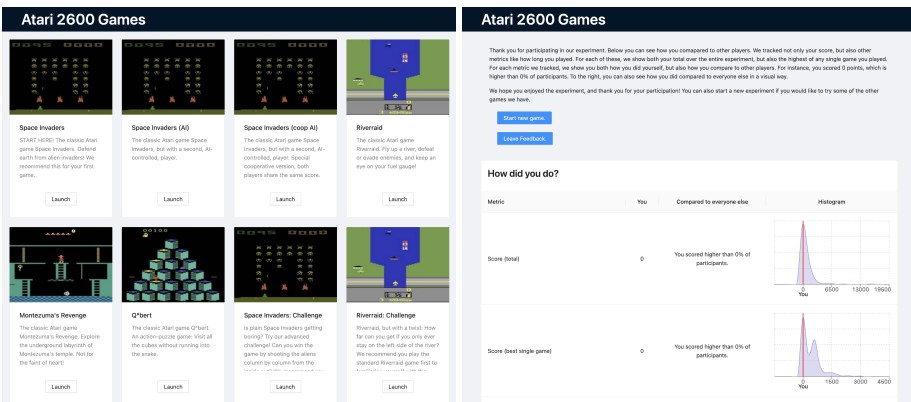

Figure 7: Screenshots of the "game choice" (left) and "high score" (right) screens of CrowdPlay.

took over control. This is to allow the remaining MTurk worker to complete their task. Similarly, if an MTurk worker was waiting for 15 minutes without a second human player connecting, a game would start with an AI agent controlling the other player. The dataset includes a record of what type of agent was controlling each player at every frame.

**AI Agent Training**    AI agents were trained in two-agent environments with two separate A2C policies. We used RLLib's implementation of A2C (Mnih et al., 2016; Liang et al., 2018) for 40 million timesteps, using standard Deepmind-style observation processing. We used Ray / RLLib version 1.4.0, and used the default hyperparameters for A2C therein. The only hyperparameters we explicitly set is the learning rate schedule: 0.0007 at training start, 1e-12 at training end, linear in between. We trained separate agents for the standard and cooperative variants as described above. Table 7 (last section) shows the hyperparameters used.

Following findings in (Tylkin et al., 2021) we trained AI agents using randomized starting positions each episode for better exploration of the states likely encountered in human-AI environments.

### A.2.7    DATASET ERRATA

In a number of space_invaders_right trajectories, we had initially set a requirement of at least 90% time spent on the right side of the screen. This proved too hard for some MTurkers, and we thus lowered the requirement to 80%. This is not distinguished through a separate task ID. MTurkers were compensated for their time in these instances.

### A.3    RECRUITMENT AND PLATFORM INTEGRATIONS

We currently target and explicitly support the following platforms.

**Amazon Mechanical Turk**    CrowdPlay can capture and store worker IDs and other metadata for payment processing. Participants are paid a fixed sum for their participation subject to minimum requirements, as well as a variable bonus payment depending on their performance. As above, these are shown to users in real-time, and stored for simple payment processing. In multi-player environments, AI agents can take over if one user disconnects, to allow remaining users to successfully complete their tasks. Participants can be assigned to tasks automatically based on previous participant's performance in a way that aims to balance the amount of high-quality data recorded for each task against defined targets.

**Direct User Payments and Raffles**    Our architecture is able to capture user data like email addresses to contact them for payment or to enter them into a raffle. Task-specific minimum time and effort requirements can be enforced similarly to MTurk.

**Lab in the Wild** (Reinecke & Gajos, 2015) is a platform for recruiting diverse participants for online experiments. Participants are not monetarily compensated, but are instead motivated through gaining insights about themselves through the experiments. Experiments are thus required to give users feedback with interesting conclusions about the user's performance and behavior. CrowdPlay supports this through its real-time statistics engine, which is used to show users a feedback page after they finish the experiment. Users are shown a variety of statistics about their gameplay behavior beyond score, e.g. shot accuracy, including a comparison with the experiment cohort. Figure 7 (right) shows a screenshot of this.

**Social Media** Similarly to Lab in the Wild, participants recruited from social media platforms are not compensated for their time but intrinsically motivated. CrowdPlay supports user choice of individual games and tasks through a modern UI resembling a "game library", and uses real-time statistics to restate task instructions as "challenges." Figure 7 (left) shows a screenshot of the "game library" feature.

### A.4 INCENTIVE EXPERIMENT TREATMENTS

We list here with more detail the five different treatments we used in the experiment in Section 4.

1. No incentives: In this treatment, we paid participants a lump sum after 5 minutes of time of running the game, irrespective of their actions or performance in the game.
2. Active Time: As above, however we only count time if there was at least one user keypress in the last 30 seconds, and at least one non-zero game reward in the last 60 seconds.
3. Minimum Requirement: In addition to 5 minutes of play time, we require that participants follow instructions to at least some minimal degree. We require they shoot a minimum of 50 aliens in the correct order, and that at least 80% of aliens shot were in the correct order (i.e. from the innermost columns that still had any aliens present).
4. Sliding Bonus: We pay participants a much smaller lump sum "base payment" after 5 minutes of active playtime and a bonus payment that reaches a maximum if participants shot at least 200 aliens in the correct order, and 100% of aliens short were in the correct order. The base and full bonus payment were chosen to equal the lump sum payment of the previous incentive regimes. The bonus payment $p$ scaled linearly in both the number of aliens shot in the correct order $n_{\text{correct}}$, as well as the fraction of aliens shot in the correct order above one-half, i.e. $n_{\text{correct}}/n_{\text{total}} - 0.5$:

$$p = p_{\max} \times \max(1, \frac{n_{\text{correct}}}{500}) \times \min(0, 2(\frac{n_{\text{correct}}}{n_{\text{total}}} - 0.5)). \tag{1}$$

5. All Incentives: We pay a small lump sum subject to the conditions in the minimum requirements regime and a bonus as in the sliding bonus regime.

In addition, we compared to students recruited by email, who were able to enter a raffle subject to a minimum performance requirement as in the treatment above; and to social media users, who we did not incentivize externally. Instead, the task was described as a "challenge" to social media users.

### A.5 INCENTIVE EXPERIMENT DATA ANALYSIS

Table 3 and Table 4 show the $p$ values for the hypothesis that the mean of the corresponding two distributions on total and fraction of high-quality data differs for each treatment pair in the incentives experiment. We denote in bold any $p$ value below .05. A value of $0.000$ denotes a $p$ value below .0005.

### A.6 T-SNE EMBEDDINGS

For the analysis performed in this paper, the t-SNE embeddings were generated using scikit-learn version 1.0.1, with PCA initialization, and all other parameters left at default. Table 7 shows the parameters used.

For the embeddings in Section 3, data was not filtered for quality, and the features used for the t-SNE algorithm were the frequency of each action per episode (six features in total).

Table 3: $p$ values for "total" measurements in incentive experiment

| vs | Email Raffle | Social Media Users | All In-centives | Scaling Bonus | Quality Re-quire-ment | Active Time |
|---|---|---|---|---|---|---|
| No incentives | **0.000** | 0.600 | **0.000** | **0.001** | **0.006** | 0.621 |
| Active Time | **0.000** | 0.362 | **0.000** | **0.000** | **0.006** | |
| Quality Requirement | 0.738 | 0.368 | 0.811 | 0.105 | | |
| Scaling Bonus | 0.341 | 0.256 | **0.021** | | | |
| All Incentives | 0.658 | 0.135 | | | | |
| Social Media Users | **0.002** | | | | | |

Table 4: $p$ values for "fraction" measurements in incentive experiment

| vs | Email Raffle | Social Media Users | All In-centives | Scaling Bonus | Quality Re-quire-ment | Active Time |
|---|---|---|---|---|---|---|
| No incentives | **0.001** | **0.000** | **0.000** | **0.001** | **0.000** | 0.440 |
| Active Time | **0.000** | **0.000** | **0.000** | **0.000** | **0.000** | |
| Quality Requirement | 0.152 | **0.039** | 0.433 | **0.035** | | |
| Scaling Bonus | 0.086 | **0.031** | **0.006** | | | |
| All Incentives | 0.123 | **0.016** | | | | |
| Social Media Users | 0.318 | | | | | |

For the embeddings in Section 5, the human data was filtered for quality using a condition of at least 80% of time spent on correct side of screen, respectively 80% of aliens shot in correct order. For the AI data, BC agents were trained using the hyperparameters described in the following section; four agents were trained using different random seeds; and each agent was used to generate 20 trajectories. The features used were action distribution as well as five features of behavioral statistics relevant to the multimodal behavior types (fraction of time spent on left / right side of screen and fraction of aliens shot in inside-out / outside-in / row-by-row order).

All t-SNE figures Section 3 and Section 5 were done using separate t-SNE embeddings for each figure. For Figure 6, it is also possible to compute a single joint t-SNE embedding for both human and BC agents. We show this in Figure 8. The separate embeddings in the main text show that t-SNE picks out different (non-"modality of behavior") features for BC data than it does for the clearly behaviorally clustered human data. On the other hand, a joint embedding shows that BC behavior is still somewhat similar (mapped to similar regions) as human behavior, albeit still significantly less clearly clustered.

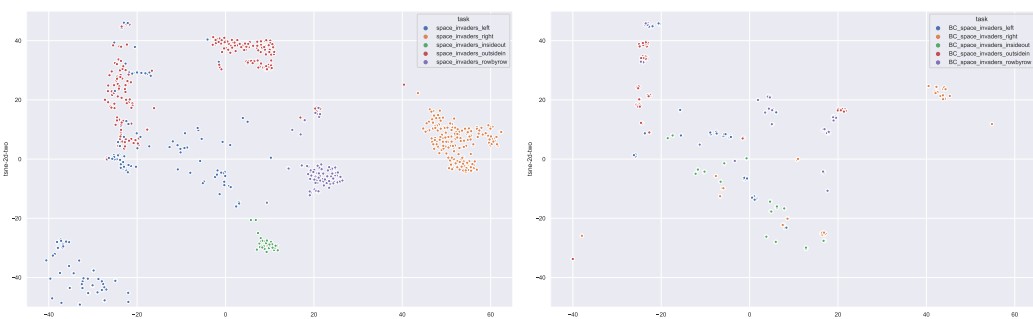

Figure 8: Joint t-SNE embedding of action distributions and behavioral statistics in multimodal behaviors for human participants (left) and BC agents (right).

A.7 BENCHMARK EXPERIMENT DETAILS

We performed all the experiments using the open source implementations of the algorithms that are provided as part of d3rlpy[2] library. We run each algorithm for 1M iterations before reporting the score. For all algorithms, we set hyper-param *n_frames* = 4 to match the framestacking of 4 frames and scaler to 'pixel'. To match the speed of evaluation environment, we downsampled the collected data to only consider every 4th transitions from the collected transitions. This is same as the standardized frame skipping operation done with Atari datasets. To avoid unfair disadvantage to offline RL algorithms due to removal of three quarters of data as a result of this downsampling, we also implemented a data augmentation technique. Here, we downsampled the data 4 times by offsetting transitions by 1 step. We used all four trajectories generated in this way for each original trajectory, thereby maintaining the total number of transitions in the original dataset. We trained each algorithm for 4 seeds and all algorithms were trained using Adam optimizer. Table 7 outlines other algorithm-specific hyper-parameter values. We used the same set of hyper-parameters as described in Gulcehre et al. (2020) to pursue a fair comparison between the performance of algorithms on human data in our paper with the performance on synthetic data reported in their paper. For metrics, we used *td_error_scorer* provided by the library. We used 'mean', 'qr' and 'iqn' as *q_func_factory* for BCQ, CQL and IQN algorithms respectively. For the normalized score, we consider best score as the one obtained by a human on the particular task. Random score is the average over 5 runs of random policy for each task and we compute the normalized score as: Normalized Score $= 100 * \frac{AlgoScore - RandomScore}{BestHumanScore - RandomScore}$. Table 5 and 6 provide individual unnormalized returns and normalized scores for individual tasks and algorithms. Figure 10 provide in-depth performance comparisons between algorithms across a suite of robust metrics. Here, Inter Quartile Mean, a statistically efficient alternative to median, discards the bottom and top 25% of the runs and calculates the mean score of the remaining 50% runs and interpolates between mean and median across runs. Optimality gap, a robust alternative to mean, represents the amount by which the algorithm fails to meet a minimum score of a threshold (1.0 in case of human performance), beyond which improvements are not very important. Agarwal et al. (2021) discusses these metrics in great details. Figure 9 displays performance profiles of each algorithm on individual tasks. The results demonstrate that IQN exhibits an overall higher performance consistently compared to other algorithms but fails to outperform humans in most tasks with couple of exceptions. DQN is comes out as a strong offline baseline while conservative approaches such as BCQ and CQL does not perform well on our dataset. The results demonstrate the difficulties for offline learning algorithms to cope with the variations in the human demonstrations and opens avenues for progress in offline learning algorithms.

Table 5: Mean Unnormalized return for offline RL algorithms. SI - Space Invaders and RR - River Raid. Scores are obtained via online evaluation. Numbers in parenthesis correspond to the best reward achieved by human for the respective task.

| Tasks | BC | BCQ | CQL | IQN | DQN | SAC |
|---|---|---|---|---|---|---|
| SI Left (1010.00) | 296.00 | 418.88 | 549.5 | 584.25 | 607.88 | 507.63 |
| SI Right (1980.00) | 381.88 | 342.00 | 363 | 642.50 | 582.13 | 573.00 |
| SI insideout (695.00) | 381.88 | 406.38 | 314.5 | 576.75 | 524.00 | 589.50 |
| SI outsidein (995.00) | 278.00 | 304.63 | 282.38 | 662.25 | 543.88 | 557.75 |
| SI rowbyrow (570.00) | 277.13 | 305.5 | 168.13 | 582.88 | 527.13 | 501.88 |
| SI vanilla (1230.00) | 294.13 | 375.75 | 395.75 | 616.75 | 601.75 | 516.00 |
| RR left (5130.00) | 1306.75 | 1406.25 | 2454.5 | 1572.00 | 1704.00 | 1058.50 |
| RR right (2070.00) | 1208.00 | 1410.75 | 1869.5 | 2054.50 | 1695.00 | 1171.25 |
| RR vanilla (3880.00) | 1372.75 | 1507.75 | 2550 | 1810.25 | 1900.25 | 1378.25 |
| Q*Bert (14950.0) | 0.00 | 168.75 | 403.75 | 1721.25 | 835.63 | 2196.25 |
| BeamRider (3932.0) | 560.00 | 618.6 | 712.75 | 1006.70 | 847.30 | 900.20 |

---

[2]https://github.com/takuseno/d3rlpy, commit hash: e89445956d8570102a8ca3b34bf0db4538b3b43e

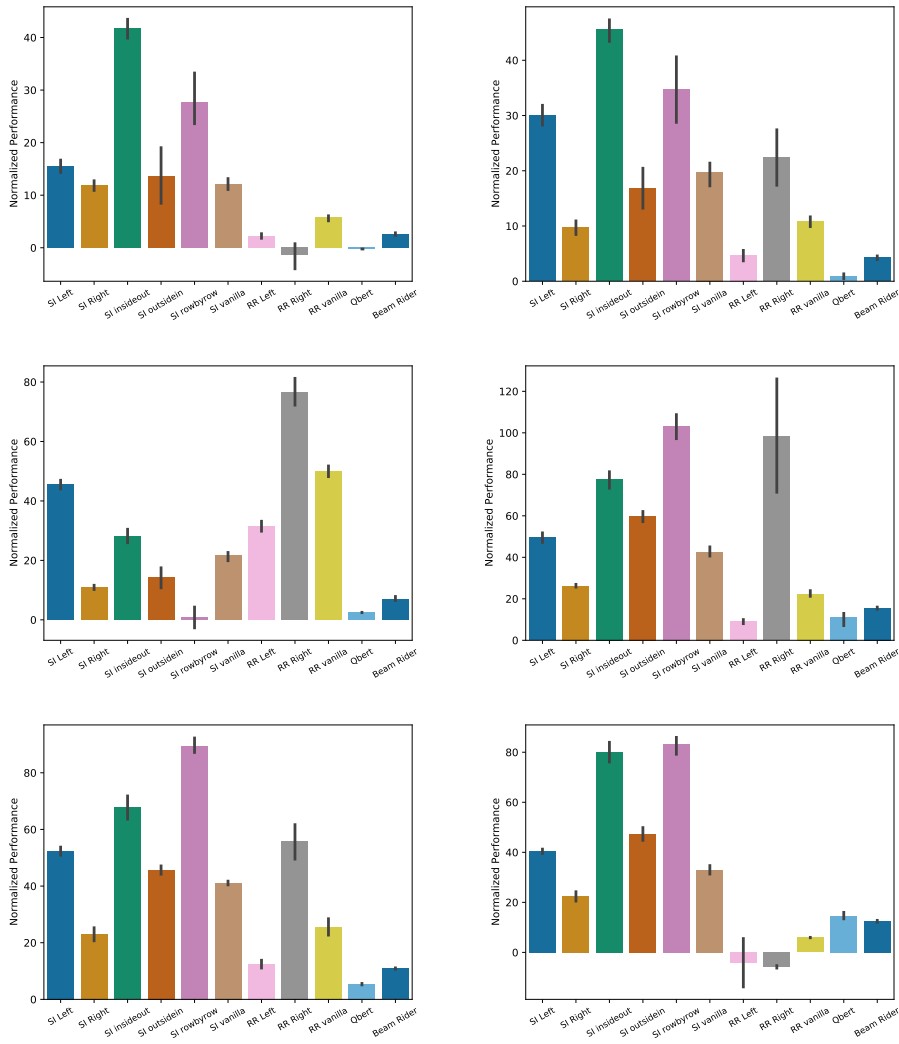

Figure 9: Task specific performance profiles for offline RL algorithms on human demonstrations. (Top-left) Performance for BC, (Top-Right) Performance for BCQ; (Center-Left) Performance for CQL, (Center-Right) Performance for IQN; (Bottom-Left) Performance for DQN, (Bottom-Right) Performance for SAC. The bars indicate the mean normalized score across seeds, and the error bars show a bootstrapped estimate of the $[25, 75]$ percentile interval for the mean estimate.

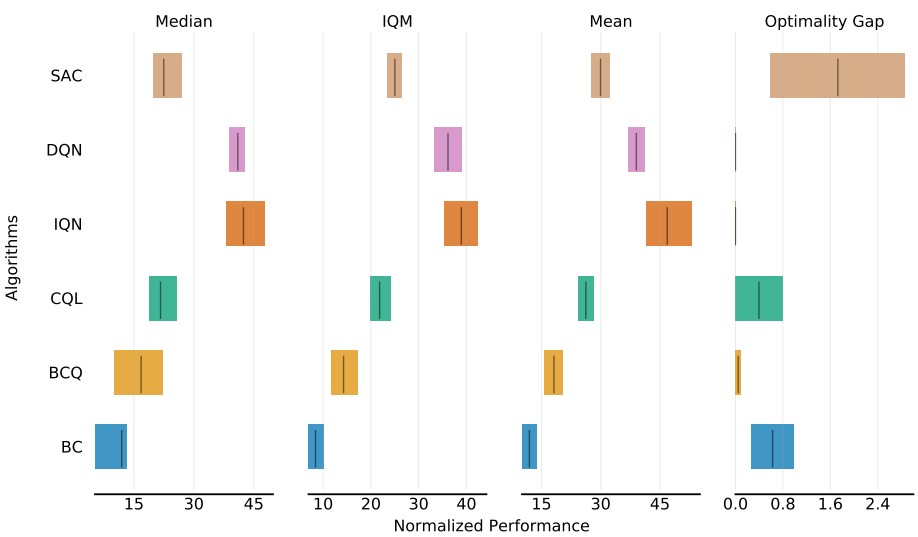

Figure 10: Aggregate metrics for offline RL algorithms on human demonstrations.).

Table 6: Mean Normalized scores for offline RL algorithms. SI - Space Invaders and RR - River Raid. Human performance score used to compute the normalized scores is available in parenthesis in Table 3. Random policy returns used to compute normalized scores are: 165.00 (shared across SI tasks), 1220.00 (shared across RR tasks), 35.00 (Qbert) and 470.40 (Beam Rider).

| Tasks | BC | BCQ | CQL | IQN | DQN | SAC |
|---|---|---|---|---|---|---|
| SI Left | 15.50 | 30.04 | 45.50 | 49.62 | 52.41 | 40.55 |
| SI Right | 11.94 | 9.75 | 10.91 | 26.30 | 22.98 | 22.48 |
| SI insideout | 41.67 | 45.54 | 28.21 | 77.69 | 67.74 | 80.09 |
| SI outsidein | 13.61 | 16.82 | 14.14 | 59.90 | 45.65 | 47.32 |
| SI rowbyrow | 27.69 | 34.69 | 0.77 | 103.17 | 89.41 | 83.18 |
| SI vanilla | 12.12 | 19.79 | 21.67 | 42.42 | 41.01 | 32.96 |
| RR left | 2.22 | 4.76 | 31.57 | 9.00 | 12.38 | -4.13 |
| RR right | -1.41 | 22.44 | 76.41 | 98.18 | 55.88 | -5.74 |
| RR vanilla | 5.74 | 10.82 | 50.00 | 22.19 | 25.57 | 5.95 |
| Q*Bert | -0.23 | 0.90 | 2.47 | 11.31 | 5.37 | 14.49 |
| BeamRider | 2.59 | 4.28 | 7.00 | 15.49 | 10.89 | 12.42 |

Table 7: Hyper-Parameter Configuration Table

| DiscreteBC | | | |
|---|---|---|---|
| **HyperParameters** | **Value** | **HyperParameters** | **Value** |
| learning_rate | 1e-3 | beta | 0.5 |
| batch_size | 100 | | |

| DiscreteBCQ | | | |
|---|---|---|---|
| learning_rate | 6.25e-5 | batch_size | 32 |
| gamma | 0.99 | n_critics | 1 |
| target_reduction_type | "min" | action_flexibility | 0.3 |
| beta | 0.5 | target_update_interval | 8000 |

| DiscreteCQL & DiscreteIQN | | | |
|---|---|---|---|
| learning_rate | 6.25e-5 | batch_size | 32 |
| gamma | 0.99 | n_critics | 1 |
| target_reduction_type | "min" | alpha | 1.0 |
| target_update_interval | 8000 | | |

| DQN | | | |
|---|---|---|---|
| learning_rate | 6.25e-5 | batch_size | 32 |
| gamma | 0.99 | n_critics | 1 |
| target_reduction_type | "min" | target_update_interval | 8000 |

| DiscreteSAC | | | |
|---|---|---|---|
| actor_lr | 3e-4 | critic_lr | 3e-4 |
| temp_lr | 3e-4 | batch_size | 64 |
| gamma | 0.99 | n_critics | 2 |
| initial_temperature | 1.0 | target_update_interval | 8000 |

| t-SNE embeddings | | | |
|---|---|---|---|
| n_components | 2 | perplexity | 30.0 |
| early_exaggeration | 12.0 | learning_rate | 200.0 |
| n_iter | 1000 | n_iter_without_progress | 300 |
| min_grad_norm | 1e-7 | metric | euclidian |
| init | pca | method | barnes_hut |
| angle | 0.5 | | |

| A2C for AI agent training | | | |
|---|---|---|---|
| gamma | 0.99 | start_lr | 0.0007 |
| end_lr | 1e-12 | learning rate schedule | linear |
| num_worker_threads | 33 | batch size | 200 |
| rollout fragment length | 20 | gradient clip | 40 |
| value function loss coeff | 0.5 | entropy coeff | 0.01 |
| gae | true | lambda | 1 |

