# OpenReview forum: "CrowdPlay: Crowdsourcing Human Demonstrations for Offline Learning"
_ICLR.cc/2022/Conference — ICLR 2022 Poster_

### Official Review · Reviewer_1Le5 · 2021-10-25

**Correctness:** 3
**Technical Novelty And Significance:** 4
**Empirical Novelty And Significance:** 3
**Recommendation:** 6
**Confidence:** 4

**Main Review:**

This paper proposes a novel framework CrowdPlay for crowdsourcing human data based on standard RL environments. The presentation of this paper is easy to follow and its content is technically sound. The design and technical details of the software structure are well disclosed and reproducible. Furthermore, the dataset provided by this paper is conducive to the research on Reinforcement Learning.

There are several issues that could be further addressed:
- Since some of the datasets provided in this paper are imbalanced, will the performance of the AI agents trained by these data be affected by the imbalance? Furthermore, more details of the AI agent training procedure should be demonstrated. E.g., how are the agents trained, and what are the differences between the standard and cooperative variants?
- As shown in Figure 3, the authors also compared the data from social media users and email raffle. What’s the incentive mechanism used on social media users?
- There are some minor issues on misspelling, e.g. "what ALE and Gym what ALE and AI Gym" (in Page. 1), “(in both Space Invaders and Riverraid,” (missing a character, in Page. 5).


**Summary Of The Paper:**

This paper proposes a novel framework CrowdPlay for crowdsourcing human data based on standard RL environments. This CrowdPlay pipeline not only supports recruiting different users from different channels to collect multimodal behaviors and data but also designs diverse and real-time incentive mechanisms to guarantee and improve the quality of data. Furthermore, the authors present a dataset, which is publicly available, along with benchmarks on Atari 2600 games, to enable further research on Imitation Learning and Offline Learning.

**Summary Of The Review:**

Overall, this is a valuable piece of work. The proposed framework CrowdPlay, including the software and the dataset, is interesting and attractive for researchers in the areas of Reinforcement Learning. It would be better if the above issues could be further addressed to make it smoother.

---

> ### Author Response · Authors · 2021-11-11
> **Response to Reviewer 1Le5**
>
> Dear Reviewer,
>
> Thank you for your review, and insightful questions. We are glad that you find our platform valuable to the community. Below we provide our response to your specific question:
>
> > Since some of the datasets provided in this paper are imbalanced, will the performance of the AI agents trained by these data be affected by the imbalance?
> - While the performance of AI agents may or may not get affected by the imbalance depending on the task, (1) We believe that in any downstream learning pipeline one would filter for data quality, and depending on the exact criteria, the balance of data between different groups might shift considerably. For instance, we have more data for the Space Invaders “Right” behavior type than for the “Left” type. But this is partly because this behavior is harder to achieve (you start on the left and have to move to the right first without getting hit by alien fire). We observe that some MTurkers played longer on the “right” task, possibly, in order to achieve the maximum payment possible, while they achieved this relatively quickly in the left type. (2) Unless the downstream research question specifically focuses on class imbalance, one way to avoid imbalance is to collect more data for a particular underrepresented class. A key advantage of CrowdPlay is the ability to do this quickly, easily, and iteratively. This also enables other groups to easily extend the existing datasets with additional data tailored to their needs.
>
> > Furthermore, more details of the AI agent training procedure should be demonstrated. E.g., how are the agents trained, and what are the differences between the standard and cooperative variants?
> - Agents were trained using A2C as implemented in RLLib, with default hyperparameters and model choice. In the updated version of the paper, we will clarify this in the text and add a table with detailed training settings to the Appendix. The difference between the standard and cooperative variants are two: (1) In the standard variant there is a bonus score when the other player gets hit by alien fire. We remove this score bonus in the cooperative variant, in order to remove incentives to hurt the other player. (2) We give both players the average of their clipped rewards instead of the usual rewards. For instance if one player hits an alien and gets an unclipped reward of 15, while the other player does not get a reward in the frame, we first clip the rewards (from 15 and 0 to 1 and 0, respectively), and then average the clipped rewards to give each agent a reward of 0.5. In the standard non-cooperative variant, the players would receive rewards of 1 and 0, respectively.
>
> > What’s the incentive mechanism used on social media users?
> - We did not give social media users any monetary incentives. For the experiment in Fig 3, we presented the “inside out” task to social media users as a “challenge” that would put a twist on the usual Space Invaders gameplay. The fact that these users gave the highest quality data suggests that intrinsic motivation beats monetary rewards in this regard, at least for the Atari games we collected data for. As noted in our response to the reviewer YicS regarding recruitment procedure, we find that recruitment on MTurk was more easily scalable compared to social media.
>
> > Misspellings
> - We have fixed the misspellings which will be available in the updated version - thank you!

---

### Official Review · Reviewer_mjxC · 2021-10-28

**Correctness:** 3
**Technical Novelty And Significance:** 3
**Empirical Novelty And Significance:** 3
**Recommendation:** 5
**Confidence:** 4

**Main Review:**

### Pros

- The paper introduces a useful tool for the community that can impact the development of the field.
- The dataset is pretty big.
- The paper provides a lot of useful information about the data collection process itself.

### Cons

The paper focuses too much on the details of data collection and the tool itself rather than providing insights about the data itself and justifying why this particular dataset can be useful to the community. In this form, this is the paper about how to effectively collect a dataset rather than a dataset itself. To make the paper impactful, I encourage the authors to focus on two points:

- What is so special about the data? What properties does it have? Can you provide any insights into the data?
- What are the challenges that your dataset poses to the community? Why should we focus on Atari? Why are the existing datasets not enough to develop strong offline RL algorithms? Section 5 of the paper should be expanded to address these questions.

Other comments/questions:

- "CrowdPlay" is also the first dataset based on Atari 2600 games that features multimodal and multiagent behaviour." Please, define multimodality a bit more formally. Atari Grand Challenge dataset you cite also has multimodal behaviour because different people might play in a different way.
- While your tool is more general, Atari Grand Challenge you present in related work also released a tool for collecting the data for the Atari games, not the dataset alone. Please, reflect this in the related work section.
- page 6: "including data such as playtime, score, and various in-game behaviour characteristics". Can you, please, specify what these various characteristics are?
- "Deepmind-style" observation processing function has to be defined more formally. Which paper it was first used in? Please, add a reference.
- A lot of software engineering/data processing details can safely go to the appendix, whereas dataset information that is currently in the appendix should be in the main body of the paper.
- Figure 3 uses the term "good data" that uses the term "task adherence". Can you, please, explain what task adherence is?
- "Our framework provides a variety of data compositions that exhibit real-world intricacies and diversities". What do you precisely mean by this? What real-world characteristics can we find in a dataset of Atari screenshots?
- "allows easily accessible crowdsourcing for any existing RL environments". Not all RL environments are gym environments.
- Can you explain the reasons behind multiplayer data? Why do we need such a dataset?
- The paper has to have hyperparameters for the experiments in Section 5. Saying 'default parameter values from d3rlpy' is not enough since the values might change over time (providing commit hash will be much better), or the repo might get removed.

**Summary Of The Paper:**

The paper introduces a tool to crowdsource data collection using a client-server architecture running OpenAI gym on the backend and providing a web interface for human players to use. Along with that, the authors release a large dataset of six Atari games, including multimodal and multiplayer data.

**Summary Of The Review:**

The paper introduces a useful tool that can be impactful for the community (Imitation Learning, Offline RL), however, I believe that it is not ready for publication due to the presentation and lack of dataset analysis. I encourage the authors to focus on the following two points:

- Describing the dataset in more detail providing the insights about the data and their properties.
- Explaining why the dataset is challenging to the existing methods, and how it can drive the further development of the fields of imitation learning and offline RL.

---

> ### Author Response · Authors · 2021-11-11
> **Response to reviewer mjxC**
>
> Dear Reviewer,
>
> Thank you for your detailed review and constructive suggestions.
> The main contributions of this  paper include both the platform---“how to collect a dataset”---and the dataset, which is one of the largest available human behavioral datasets, to the best of our knowledge. We do not know of a comparable platform, and as a result the area of offline learning is underserved, as also noted by reviewer YicS. Learning from human data has been critical in advancing many other areas of ML, but much less the case for RL and partly due to difficulties with such data collection. By providing the first human behavior dataset at this scale, and a ready methodology to collect more data, we hope to change this. The flexibility of the CrowdPlay pipeline enables researchers to collect any data (not specific to Atari or even Gym as discussed below) they want, and with the benefits of real-time incentives and task design (multimodality) possibilities as noted by reviewer 1Le5.
>
> We will add more details on the dataset to the paper and we welcome the reviewer’s suggestion on any specific analyses or experiments to address any concerns that are not covered in our responses below and would help to convince them further of the value of this work.
>
> We will address your general points and your specific questions in a separate comments.

---

> > ### Author Response · Authors · 2021-11-11
> > **Addressing your general points**
> >
> > > What is so special about the data?
> > - The dataset is by far the largest dataset of human Atari gameplay to date, and the only dataset collected in real-time and using the ALE emulator. That in itself makes it interesting. In addition, it features explicit multimodal behavior, a multiagent dataset with several interesting variations (human-AI vs human-human, cooperative vs competitive), and rich metadata and gameplay statistics, all of which has a lot of value in the field of offline reinforcement learning. Because this is human behavior data, it has properties mimicking real-world complexity, including noise and variation in behavior.
> >
> > >What properties does it have? Can you provide any insights into the data?
> >
> > - We will add a more detailed discussion of descriptive statistics such as distribution of trajectory length, scores, or actions. For instance, we can analyze the action distribution of both human and AI players in different tasks. A t-SNE embedding of action distributions from 100 episodes of both standard and cooperative human-AI Space Invaders shows a clear distinction between human and AI action distributions, two clear clusters in AI distributions corresponding to cooperative and competitive behavior, and a clear but involved relationship between cooperative/competitive variants and action distribution in the human data. The following figure shows this: [T-SNE embedding of action distributions](https://drive.google.com/uc?export=view&id=1qi2IlT7_xWHU5lPeWNkWyqCFSS6hGljT)
> > Is this the type of analysis you had in mind? Please also let us know if there are any other specific properties or insights and we will be happy to incorporate them.
> >
> > >What are the challenges that your dataset poses to the community?
> > - The challenge comes from this being human behavioral data, thus showing different characteristics than synthetic datasets.  A key role of an offline RL and IL dataset is to present interesting behavior data to advance the design of robust and effective algorithms.  As we note above, this data has properties mimicking real-world complexity, including noise and variation in behavior. Beyond offline RL and IL,  tasks such as behavior classification, intent recognition, and human-AI interaction can also benefit from this kind of data. See also below for why existing datasets do not pose these same challenges.
> >
> > >Why should we focus on Atari?
> > - Atari is an interesting challenge for offline learning for the same reasons it has been pivotal for online learning: it is designed for human players and has subtleties and quirks not found in made-for-purpose simulators, capturing a semblance of real-world complexity. At the same time, it is just about within reach of current state-of-art-methods. It has also been shown to be interesting for investigating human learning [1]. Having noted that, we emphasize that CrowdPlay is not Atari specific and supports any other gym-like environment.
> >
> > > Why are the existing datasets not enough to develop strong offline RL algorithms?
> > - As far as human behavior data on Atari is concerned, existing datasets are problematic for offline learning applications. The Atari Head dataset was recorded in a semi frame-by-frame mode and is not representative of normal human gameplay. Atari Grand Challenge uses Javatari, a different simulator than that commonly used for learning (ALE) that cannot be guaranteed to have identical behavior and thus state transitions. For example, we are unable to reproduce the Atari Grand Challenge trajectories in ALE (by feeding action sequences into ALE), and were advised by the Javatari developer  against using Javatari for this purpose. Aside from these issues, neither of these datasets match ours in either volume or breadth and we are not aware of similarly large human datasets on other domains.
> > - There are synthetic datasets available, such as RL Unplugged and D4RL (this latter dataset includes human data on a few tasks, but is largely synthetic). These serve a  different purpose than human datasets, as they do not capture the noise and variance inherent in human behavior. Indeed, the RL Unplugged paper [2] benchmarks the same algorithms we consider on synthetic Atari data and finds that these perform well (Fig 6 on page 9). A detailed per-game evaluation in Table 9 and Figure 7 (p17-18 therein) also shows this to be the case specifically for the games that we benchmark, where these same algorithms fail on human data. As we already filter for data quality in our algorithm benchmarks, this clearly demonstrates that more work is needed for learning from human data. We will add this discussion to the paper.

---

> > > ### Author Response · Authors · 2021-11-11
> > > **Addressing Specific Questions (Part 1/2)**
> > >
> > > > Please, define multimodality a bit more formally.
> > > - We mean here explicit multimodality in the sense that participants were asked to follow specific behavior types. Additionally, we provide quantitative measurement of how closely the behavior follows each of these types, including a way to apply this same measurement to AI-generated trajectories; e.g., for evaluation of multimodal imitation learning methods in downstream learning pipelines. We will clarify this in the paper.
> > >
> > > > Atari Grand Challenge you present in related work also released a tool for collecting the data for the Atari games
> > > - As noted above, unlike our platform, Atari Grand Challenge is specific to Atari games, and as per above, it uses a different simulator (Javatari [5]) than the standard ALE [6] used in learning pipelines. We will add the public availability of their platform in our related section, thank you for pointing that out.
> > >
> > > > Can you, please, specify what these various characteristics are?
> > > - These are configured differently for each task and can be any measure of behavior type, for instance: in a multimodal task where we ask the user to always keep their spaceship on the left half of the screen, we record and display in real time to the user what fraction of time they have in fact been spending on the left half of the game screen. In another task, we measured the fraction of alien invaders shot in a specific order. We will clarify this in the paper.
> > >
> > > > "Deepmind-style" observation processing function has to be defined more formally
> > > - We refer here to standard preprocessors such as frameskipping, taking the pixel-wise maximum of two consecutive frames to account for Atari alternate-frame rendering (usually implemented together as “max-and-skip”), rescaling to 84x84 grayscale images, and framestacking, which we believe in this combination first appeared in [4], although some individual parts such as frameskipping appear  earlier. Frameskipping, in particular, is a challenge in our context, as its standard implementation only shows an observation to, and samples an action from, an AI policy on every 4th frame, which we do not want to do for human players. We will add this description and related details to the paper.
> > >
> > > > A lot of software engineering/data processing details can safely go to the appendix, whereas dataset information that is currently in the appendix should be in the main body of the paper
> > > - Thanks. We will work to gain better balance in the paper presentation in the updated version.
> > >
> > > > Can you, please, explain what task adherence is?
> > > - By task adherence we mean how closely the user matched the behavior type we asked them to perform, as defined and measured by the real-time statistics we collect for each task. For instance, in one task we ask users to shoot alien invaders in a specific order. If in an episode a user shot 8 aliens in that order, but 2 aliens out of order, we would consider this a task adherence of 0.8. The metadata engine also makes it easy to modify this logic, e.g., additionally filter for a minimum episode score or other criteria.

---

> > > > ### Author Response · Authors · 2021-11-11
> > > > **Addressing Specific Questions (Part 2/2)**
> > > >
> > > > > What real-world characteristics can we find in a dataset of Atari screenshots?
> > > > - There are two ways in which this data exhibits real-world or real-world-like complexities. First, this comes through the design of the Atari games themselves, as has been argued in the RL literature (e.g., the original ALE paper). Because these environments are exogenously given and have been designed for human players in mind, they exhibit many subtle complexities and quirks not present in made-for-purpose simulators. Secondly, because our dataset is collected from human participants, it will show many of the complexities inherent to human behavior, e.g. variation between individuals, irrational behavior, etc.
> > > >
> > > > > Not all RL environments are gym environments.
> > > > - Indeed. CrowdPlay is not specific to Gym environments, which is why we state “and Gym-like environments” in the paper. The environment we used in collecting the Atari dataset is technically not a Gym environment, but adheres to the RLLib multiagent specification. Broadly speaking, our pipeline can take any object that has a reset() and a step() function, and crowdsource data for it. This includes Gym and RLLib environments, and many others. In fact, we cannot think of an RL environment that does not in some way expose state transitions through a step function. We are also not limited to the specific observation and action spaces that Atari games use.
> > > >
> > > > > Can you explain the reasons behind multiplayer data? Why do we need such a dataset?
> > > > - We believe this dataset will be useful for instance for developing and benchmarking multiagent imitation and offline learning methods. Similarly to the single-agent data, there is interesting variety here, in that the dataset has both human-AI and human-human data, and both competitive and cooperative variants of the same game. The t-SNE embedding of action distributions we included in our other comment to you shows that the different incentive structures (cooperative vs competitive rewards) affect human behavior in subtle ways, and we expect other aspects of multiagent environments (e.g. different coordination modalities) to do the same. This can be an interesting benchmark for multiagent methods that should learn these aspects of behavior. The pipeline itself can also be useful beyond data collection for the development of offline RL methods, for instance in human-AI interaction research. The platform enables human-AI experiments with minimal effort, allowing researchers to go directly from training an AI agent in a standard multiagent RL environment to testing the AI’s interaction with people.
> > > >
> > > > > The paper has to have hyperparameters for the experiments in Section 5.
> > > > - Thank you for pointing this out, we will add details on the hyperparameters used as well as the commit hash (that is a great suggestion) in the updated version.
> > > >
> > > >
> > > > [1] Tsividis, Pedro A., et al. "Human learning in Atari." 2017 AAAI Spring Symposium Series. 2017.
> > > >
> > > > [2] Gulcehre, Caglar, et al. "RL Unplugged: A Collection of Benchmarks for Offline Reinforcement Learning." Advances in Neural Information Processing Systems 33 (2020).
> > > >
> > > > [3] Fu, Justin, et al. "D4rl: Datasets for deep data-driven reinforcement learning." arXiv preprint arXiv:2004.07219 (2020).
> > > >
> > > > [4] Mnih, Volodymyr, et al. "Human-level control through deep reinforcement learning." Nature 518.7540 (2015): 529-533.
> > > >
> > > > [5] https://github.com/ppeccin/javatari
> > > >
> > > > [6] Machado, Marlos C., et al. "Revisiting the arcade learning environment: Evaluation protocols and open problems for general agents." Journal of Artificial Intelligence Research 61 (2018): 523-562.

---

> > > > > ### Comment · Reviewer_mjxC · 2021-11-19
> > > > > **response**
> > > > >
> > > > > Thanks a lot for your updates! The draft looks better right now and I'm increasing my score which was 3 before the update. While I believe that the paper has great potential for the community, I still think it requires more work.
> > > > >
> > > > > I still believe that the paper has too much information on 'how to collect the data' rather than showing the benefits of the data. Maybe the 'additional experiments' mentioned in [this comment](https://openreview.net/forum?id=qyTBxTztIpQ&noteId=lvY5hVhdS3U) will clarify it better.
> > > > >
> > > > > It's great that the authors notice the difference between the performance of the algorithms on synthetic data and on human data. However, the most important question is "What are the challenges the human data poses", right now it's unclear. Moreover, I'm not sure that comparison to the RL Unplugged is apples to apples comparison. Comparing, for instance, behavioural cloning hyperparameters, RL Unplugged uses a smaller learning rate and a much larger batch size (1000 vs your 100). I guess this might be another reason for the inconsistency in the performance.
> > > > >
> > > > > Minor comments:
> > > > > * You mention that you use default PCA parameters from scikit. The parameters might change in the library in the future, it is better to give exact values.
> > > > > * t-SNE plot labels are too small and hard to read.

---

> > > > > > ### Author Response · Authors · 2021-11-23
> > > > > > **Response**
> > > > > >
> > > > > > Dear Reviewer,
> > > > > >
> > > > > > thank you for giving careful consideration to our responses and revisions and for your further suggestions. We have now added additional discussion and experiments to section 5 in order to further strengthen the case that our dataset is useful, and to address concerns regarding inconsistent hyperparameters.
> > > > > > * Specifically, we have considered two algorithms BC (Behavior Cloning) and IQN (Inter-Quantile Network) that were shown to perform worst and best respectively on AI trained data in the RL unplugged paper. We re-ran our experiments with the exact hyper-parameters used in the RL unplugged paper, evaluated on 4 seeds per task per algorithm, and reported performance profiles using the method from Agarwal et al. as recommended in the public comment. These experiments provide further support to our initial findings as we observe that the best performing IQN on AI trained dataset does not even perform as well as BC on human demonstrations. We will add similar details on other baseline algorithms (offline DQN, CQL and BCQ) in the camera-ready version if accepted.
> > > > > > * We also qualitatively compare the style of play of both human agents and cloned agents trained using BC, on the five explicit multimodal behavior types we collected in Space Invaders. We see that BC fails to emulate human behavior as measured on even simple behavioral statistics such as time spent on left or right side of screen (where that was the objective and very clearly followed by the human participants).
> > > > > > *Additionally, we added hyperparameters for the t-SNE algorithm, and made the t-SNE figures larger. We will continue to further improve the layout of the camera-ready version if accepted.
> > > > > >
> > > > > > Please let us know if this addresses your concerns or if any further questions remain open.

---

> > > > > > > ### Comment · Reviewer_mjxC · 2021-11-25
> > > > > > > **response**
> > > > > > >
> > > > > > > Thanks for your updates. I think Figure 8 is an interesting result and encourage the authors to include it in the main body of the paper rather than hiding it in the Appendix.
> > > > > > >
> > > > > > > In my opinion, showing that offline RL methods < behaviour cloning on the human dataset will be an impactful paper in the field. However, I believe that the paper requires a little bit of work, and I have no control over paper changes if it gets accepted at this review cycle. For instance, I cannot guarantee this: "We will add similar details on other baseline algorithms (offline DQN, CQL and BCQ) in the camera-ready version if accepted."
> > > > > > >
> > > > > > > I think the paper has improved since the initial submission, however, I consider my current score to be fair.

---

> > > > > > > > ### Author Response · Authors · 2021-11-30
> > > > > > > > **Additional Results available now**
> > > > > > > >
> > > > > > > > Thank you for your response. We were unable to include all baselines by the paper update deadline due to computational constraints, but some of the remaining experiments have now finished, and a handful are still running with results coming in soon. We have uploaded updated results on the link below, and will fill in the few remaining results as they come in:
> > > > > > > > [Benchmark Performance Results](https://drive.google.com/drive/folders/1-z-pbk-2dkz4ydzgWboCnRzIhhtMuYyr?usp=sharing)
> > > > > > > >
> > > > > > > > As per your suggestion, we will add the full version of Figure 8 to the main text and reorganize the text for better presentation by moving some contents on data processing to the Appendix.
> > > > > > > >
> > > > > > > > Thank you once again for engaging in detailed discussions and providing insightful suggestions, which have been very fruitful.

---

### Official Review · Reviewer_YicS · 2021-11-06

**Correctness:** 3
**Technical Novelty And Significance:** 4
**Empirical Novelty And Significance:** 2
**Recommendation:** 8
**Confidence:** 4

**Details Of Ethics Concerns:**

Possible issues with not enough controls to ensure participants are fairly compensated and not exploited which is common in crowdsourcing platforms.

**Main Review:**

Learning from human labelled datasets is a tried and tested paradigm to make progress on hard learning problems. We’ve seen this play out in CV and NLP for long, and although some RL work does rely on human demonstration data, collecting these datasets is often an ad-hoc process relying on designing custom, specific pipelines for each problem. This paper introduces a platform that makes it seamless to crowdsource human demonstrations for any underlying simulator. The platform also provides an easy way to post process the collected data for learning experiments.

This is an important and refreshing piece of work: better research tools and infrastructure are leverage multipliers on productivity of downstream research work. The specific area: a pipeline for collecting labelled offline RL datasets also seems under-served.

I like the design choices the paper makes in building this platform and they seem to have been well thought. In particular, not requiring the simulator / emulator to be run in the browser is a big plus, which means the platform can serve a wide range of simulators via the client / server architecture. I also appreciate the authors providing an anonymous staging for the platform, and that the entire platform will be open-source.

I do have some questions for clarification:
  * Does CrowdPlay enable researchers to recruit participants directly from the platform? It doesn’t seem like it (since the data collected itself was from several channels), but if it does offer an option it would be nice to elaborate on what steps CrowdPlay takes to ensure players are fairly compensated.
  * The extend of the multi-modal dataset seems quite small, limited to 2 games. Are there plans to expand the dataset?
  * Have you bencharked the latency of the architecture (time taken to do one keypress->send-to-server->env.step()->send obs to client loop)? I would imagine high latency would disincentive participants from completing the demos.

Finally, I believe you should these two references which are related to your work: [1] shows that even unaligned YouTube videos can be leveraged for playing hard exploration games in Atari. [2] is the state-of-the-art offline RL algorithm on Atari games.

[1] Aytar, Y., Pfaff, T., Budden, D., Paine, T. L., Wang, Z., & de Freitas, N. (2018). Playing hard exploration games by watching YouTube. NeurIPS 2018.

[2]  Schrittwieser, J., Hubert, T., Mandhane, A., Barekatain, M., Antonoglou, I., & Silver, D. (2021). Online and offline reinforcement learning by planning with a learned model. NeurIPS 2021

**Summary Of The Paper:**

This paper presents CrowdPlay, a crowdsourcing platform to collect human demonstrations for any MDP. It also accompanies a dataset of human gameplay on Atari games with multi-agent and some multi-behavior aspects. The paper benchmarks existing offline RL algorithms on this dataset, and details incentive design mechanisms for future crowdsourcing jobs.

**Summary Of The Review:**

This paper presents an important and under-looked infrastructure contribution: a platform to make collecting human data easy for any existing simulators. The platform makes use of good decision decisions. The accompanying Atari dataset could be useful for useful for muli-agent and multi-modal RL research as well. Therefore I recommend a strong acceptance for this paper.

---

> ### Author Response · Authors · 2021-11-11
> **Response to Reviewer YicS**
>
> Dear Reviewer,
>
> Thank you for your supportive review and we are glad that you find our platform interesting and useful to the community. Thank you for providing helpful pointers on playing hard exploration games and offline learning. We will incorporate them in the revised version of the paper which will be updated soon. We address your specific questions below:
>
> > Does CrowdPlay enable researchers to recruit participants directly from the platform?
> - When we recruit through social media and email, we direct participants to our platform URL. They are then presented with several choices for which game and variant they would like to play. This can be considered a form of recruitment via our platform. For such recruitment, our platform also allows capturing user data such as email addresses. In our own email recruitment, we use this to compensate participants by entering them into a raffle for gift cards. For social media users, we do not provide monetary compensation, and instead build on intrinsic motivations. Please refer to our answer to reviewer 1Le5 regarding this part for more details.
> - On paid platforms like MTurk, we show workers their expected payment in real-time, allowing them to make an informed decision regarding how much time they would like to put in. We have generally found that recruitment through platforms such as MTurk is more scalable than direct recruitment such as via social media.
> - We also display a form that the user needs to accept before participation so that they are aware of the specifics of their participation.
>
> > Are there plans to expand the dataset?
> - Yes, absolutely! In fact, our dataset collection is still ongoing and the dataset now comprises well over 400 hours of data and we will add additional games and multimodal tasks in the future. One key benefit of CrowdPlay is that it is easy to collect additional data based on the evolving needs of a researcher. For instance, if there is a particular game and multi-modal behavior type of interest to the reviewer we can collect this during the discussion period.
>
> > Have you benchmarked the latency of the architecture?
> - Yes! This is something the pipeline already stores as part of the trajectory data: On every keypress, the client transmits the keypress and the sequence number of the current frame shown on screen. From this we can reconstruct the end-to-end latency (including both network and processing latency). From an analysis of 1000 randomly chosen episodes, we find an average latency of 5.75 frames or 95 ms. Given human reaction time is around 250-300ms [1], we consider this acceptable. We have received very little  feedback regarding latency problems. Our deployment on Elastic Beanstalk is designed to support multiple instances spread across geographic regions, with automatic routing of requests to the closest instance. The data can also be filtered for acceptable levels of latency if this does become  a concern.
>
> [1] Zhang, Ruohan, et al. "Atari-head: Atari human eye-tracking and demonstration dataset." Proceedings of the AAAI conference on artificial intelligence. Vol. 34. No. 04. 2020.

---

> > ### Comment · Reviewer_YicS · 2021-11-22
> > **Response**
> >
> > Thanks, it's a bit concerning that the platform is used to actively recruit participants and there are not enough controls to ensure participants are fairly compensated. There's a long history of systematic under-compensation on MTurk, and I believe new platforms should not repeat the same mistake. I am not an expert here, so I'm flagging this for ethics review.

---

> > > ### Author Response · Authors · 2021-11-23
> > > **Response**
> > >
> > > Dear Reviewer,
> > > Thank you for bringing up this concern. We had misinterpreted your original question to be about what recruitment capabilities the platform has, rather than mainly about compensation and ethics, and answered accordingly. However, we care deeply about the ethics of our work, and agree that fair compensation can be an issue. Significant effort and discussion has gone into designing our platform to be effective in this regard.  We have added a detailed ethics section to the PDF, which explains the thinking behind our platform design in this regard, as well as the remaining issues that we have found. We would like to also directly address your specific concerns:
> > >
> > > * All of our own data collection for the CrowdPlay Atari dataset underwent IRB review, including review of details on recruitment and compensation.
> > > * We have designed CrowdPlay to enable strong safeguards, in particular through its real-time statistics capability. The platform allows us to display to participants detailed, specific criteria for compensation (e.g., “hit 50 aliens”), and real-time progress toward these criteria as well as real-time expected payments. This enables fairer payments through: (a) communicating requirements and progress clearly to participants, avoiding participants putting in honest effort but failing to understand what is expected of them; (b) giving participants the opportunity to make an informed decision about their participation; and (c) creating accountability, as we commit to a compensation scheme ahead of time.
> > > * The platform allows us to implement the same controls and safeguards for direct recruitment as we do on MTurk. The only difference would be the way in which payments are processed.
> > > * In the case of students recruited through email lists, we used a raffle instead of fixed payments, based on feedback from students themselves that they preferred this over smaller guaranteed payments. The requirements to enter the raffle were still communicated using the same real-time feedback
> > > * For the Atari dataset, and specifically for this domain because it is videogames which many people enjoy playing, we also recruited unpaid participants. We weighed carefully whether this makes sense here and we are not advocating that this will be suitable in  other domains. We took care to target participants who were already looking for videogames to play, and in selecting appropriate experiments to offer  so that they were likely to be fun. We consider unpaid recruitment an exception due to the nature of our domain, not something we advocate for as a norm.
> > > * For multiagent data collection, we developed fallback AI capabilities specifically to ensure fair compensation for the remaining participant(s) if a participant leaves before the experiment is complete.
> > > * While of course for future uses of the CrowdPlay platforms the specific compensation scheme will be up to the individual researchers designing the experiment, we hope that these features will help enable better best practices in regard to fair compensation.
> > >
> > > We hope that this answers some of your concerns. Please let us know in case anything remains open. And again, thank you for bringing this up.

---

> > ### Comment · Reviewer_YicS · 2021-11-22
> > **References**
> >
> > Also, are you planning to include the references on offline learning and learning Atari games from Youtube?

---

> > > ### Author Response · Authors · 2021-11-23
> > > **Response**
> > >
> > > Yes, absolutely, our sincere apologies that we hadn’t done this yet, this was an oversight on our part. We have added the references to the paper now. We also plan to benchmark Reanalyze and MuZero Unplugged from Schrittwieser et al in the camera-ready version if the paper is accepted.

---

### Public Comment · ~Rishabh_Agarwal2 · 2021-11-10
**Advantages over existing Atari dataset & Suggestion for reliable evaluation**

Hi authors,

I didn't find this in the paper but what are the advantages of this dataset over the existing Atari dataset (DQN Replay, RL Unplugged): https://research.google/tools/datasets/dqn-replay/ which provided 200M frames of interaction of DQN agent over 50+ Atari games?

Re benchmark results: Median scores remain unchanged even if set the scores on nearly half of the task to be zero while mean score are highly prone to outliers. Instead, aggregate scores such as interquartile mean and optimality gaps across all runs and tasks (using total of 10 tasks x 5 seeds = 50 seeds) might be better to report.

Additionally, performance profiles would capture variability in performance across tasks which are not revealed by aggregate scores.  Since you have access to scores on individual seeds, you can easily incorporate the above recommendations using the library at https://github.com/google-research/rliable or the corresponding [colab](https://bit.ly/statistical_precipice_colab).

[1] Agarwal, R., Schwarzer, M., Castro, P.S., Courville, A. and Bellemare, M.G., 2021. Deep reinforcement learning at the edge of the statistical precipice. In NeurIPS. https://openreview.net/forum?id=uqv8-U4lKBe

[2] Gulcehre, Caglar, et al. "RL Unplugged: A Collection of Benchmarks for Offline Reinforcement Learning." In NeurIPS (2020)

---

> ### Author Response · Authors · 2021-11-11
> **Response to your comment**
>
> Thank you for your comment and useful suggestions, including pointing us to the work on evaluating benchmark results. We will incorporate this into the paper.
>
> Regarding RL Unplugged, the advantages of our work are twofold. (1) Our data is collected from human players and therefore shows noise and variation that is not present in synthetic data, and it is by far the largest such dataset on this domain. (2) A key contribution of our platform is that it enables crowdsourcing of human behavior data in arbitrary RL environments. This is noted, and as per our discussion with reviewer mjxC, we will add a discussion of the performance of algorithms on the RL Unplugged Atari dataset. This provides a useful counterpoint, highlighting the new difficulties faced by these offline RL algorithms on human behavioral data.

---

### Author Response · Authors · 2021-11-17
**Updated PDF**

Dear Reviewers,

We have now updated the PDF with changes to address the reviewers’ comments.
The main changes are:
* We have added a discussion of characteristics of the dataset in section 3.3 in response to reviewer mjxC’s questions about insights into the data. We analyze here the action distribution of participants recruited from different channels and show that these demographics introduce clear variance into the dataset. We also analyze the action distributions of human and AI agents in standard and cooperative two-player Space Invaders, and show differences in behavior between these.
* We have restructured the paper to put more focus on the dataset, while moving some details on data processing to the appendix, as suggested by reviewer mjxC.
* We include references to synthetic datasets and discuss RL Unplugged benchmarks in section 5.
* We added details and hyperparameters for AI training and benchmarks in the appendix.
* Some additional smaller changes (mention Atari Grand Challenge software availability, clarify “good” data in incentive experiment, clarify incentives for social media participants, fixed typos)
* We are currently working on additional experiments to extend section 5.

---

### Author Response · Authors · 2021-11-23
**Updated PDF (2)**

Dear Reviewers,

We have now uploaded a further revision of our paper as a PDF to address remaining concerns. The main changes are:
* Additional comparison of human and BC agents’ behavior in Section 5.
* Additional evaluation of performance of BC and IQN in Section 5 and the appendix.
* Addition of an ethics statement.
* Addition of references suggested by reviewer YicS
* Addition of hyperparameters for t-SNE
* Minor reformatting (larger t-SNE plots)
* Moved some details to the appendix

---

### Decision · Program_Chairs · 2022-01-20

**Decision:**

Accept (Poster)

**Comment:**

This paper studies the problem of how to collect demonstrations via crowd sourcing for imitation and offline learning. The paper received mixed reviews initially. The reviewers had difficulty understanding empirical results, asked for some more ablations, and were little unconvinced by the proposed usefulness of the collected data. The authors provided a strong thoughtful rebuttal that addressed many of those concerns. The paper was discussed extensively with one of the reviews who increased their score from 3 to 5. Reviewers generally agree that the paper is good but not all reviewers are on-board with acceptance. AC recommends accept but agrees with the reviewers and the authors are urged to look at reviewers' feedback and incorporate their comments in the camera-ready.